# OMNI-CAPTIONER: DATA PIPELINE, MODELS, AND BENCHMARK FOR OMNI DETAILED PERCEPTION

[1,2]**Ziyang Ma**,[*][1]**Ruiyang Xu**,[*][3]**Zhenghao Xing**,[*][5]**Yunfei Chu**, [5]**Yuxuan Wang**, [5]**Jinzheng He**, [5]**Jin Xu**,[†][3]**Pheng-Ann Heng**, [1]**Kai Yu**, [5]**Junyang Lin**, [2]**Eng-Siong Chng**, [1,4]**Xie Chen**[‡]

[1]Shanghai Jiao Tong University, [2]Nanyang Technological University, [3]The Chinese University of Hong Kong, [4]Shanghai Innovation Institution, [5]Qwen Team, Alibaba Group

## ABSTRACT

Fine-grained perception of multimodal information is critical for advancing human–AI interaction. With recent progress in audio–visual technologies, Omni Language Models (OLMs), capable of processing audio and video signals in parallel, have emerged as a promising paradigm for achieving richer understanding and reasoning. However, their capacity to capture and accurately describe fine-grained details remains limited explored. In this work, we present a systematic and comprehensive investigation of omni detailed perception from the perspectives of the data pipeline, models, and benchmark. We first identify an inherent "co-growth" between the level of detail and the degree of hallucination in current OLMs. To address this, we propose **Omni-Detective**, an agentic data generation pipeline integrating tool-calling, to autonomously produce highly detailed yet minimally hallucinatory multimodal data. Based on the data generated with Omni-Detective, we train two captioning models: **Audio-Captioner** for audio-only detailed perception, and **Omni-Captioner** for audio–visual detailed perception. Under the cascade evaluation protocol, Audio-Captioner achieves the best performance on MMAU and MMAR among all open-source models, surpassing Gemini 2.5 Flash and delivering performance comparable to Gemini 2.5 Pro. On existing detailed captioning benchmarks, Omni-Captioner sets a new state-of-the-art on VDC and achieves the best trade-off between detail and hallucination on the video-SALMONN 2 testset. Given the absence of a dedicated benchmark for omni detailed perception, we design **Omni-Cloze**, a novel cloze-style evaluation for detailed audio, visual, and audio-visual captioning that ensures stable, efficient, and reliable assessment. Experimental results and analysis demonstrate the effectiveness of Omni-Detective in generating high-quality detailed captions, as well as the superiority and human preference alignment of Omni-Cloze in evaluating such detailed captions. All the data pipeline, models, and the benchmark are open-source to facilitate further research for omni detailed perception. [1]

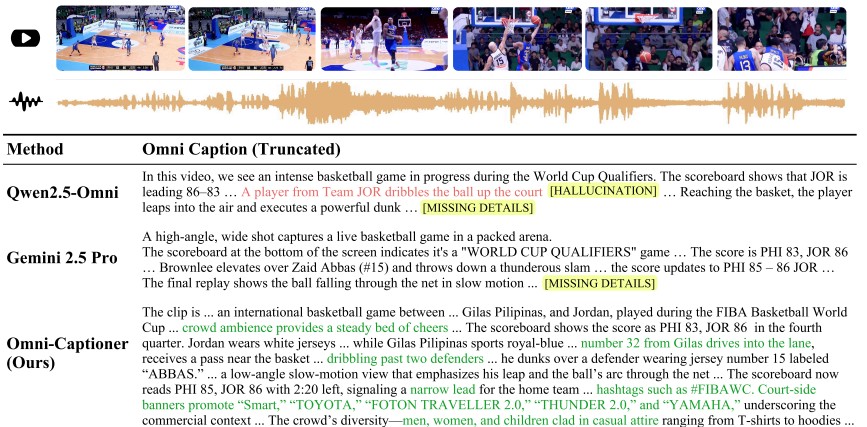

| Method | Omni Caption (Truncated) |
|---|---|
| **Qwen2.5-Omni** | In this video, we see an intense basketball game in progress during the World Cup Qualifiers. The scoreboard shows that JOR is leading 86–83 … A player from Team JOR dribbles the ball up the court [HALLUCINATION] … Reaching the basket, the player leaps into the air and executes a powerful dunk … [MISSING DETAILS] |
| **Gemini 2.5 Pro** | A high-angle, wide shot captures a live basketball game in a packed arena. The scoreboard at the bottom of the screen indicates it's a "WORLD CUP QUALIFIERS" game … The score is PHI 83, JOR 86 … Brownlee elevates over Zaid Abbas (#15) and throws down a thunderous slam … the score updates to PHI 85 – 86 JOR … The final replay shows the ball falling through the net in slow motion ... [MISSING DETAILS] |
| **Omni-Captioner (Ours)** | The clip is ... an international basketball game between ... Gilas Pilipinas, and Jordan, played during the FIBA Basketball World Cup ... crowd ambience provides a steady bed of cheers ... The scoreboard shows the score as PHI 83, JOR 86 in the fourth quarter. Jordan wears white jerseys ... while Gilas Pilipinas sports royal-blue ... number 32 from Gilas drives into the lane, receives a pass near the basket ... dribbling past two defenders ... he dunks over a defender wearing jersey number 15 labeled "ABBAS." ... a low-angle slow-motion view that emphasizes his leap and the ball's arc through the net ... The scoreboard now reads PHI 85, JOR 86 with 2:20 left, signaling a narrow lead for the home team ... hashtags such as #FIBAWC. Court-side banners promote "Smart," "TOYOTA," "FOTON TRAVELLER 2.0," "THUNDER 2.0," and "YAMAHA," underscoring the commercial context ... The crowd's diversity—men, women, and children clad in casual attire ranging from T-shirts to hoodies ... |

Figure 1: Comparison of the detailed captioning among the Omni-Captioner and other omni models.

---

[*]Equal Contribution.

[†]Project Leader.

[‡]Corresponding Author.

[1]https://github.com/ddlBoJack/Omni-Captioner

# 1 INTRODUCTION

Recent advances in Omni Language Models (OLMs) have enabled machines to produce increasingly rich descriptions of audio–visual scenes (Cheng et al., 2024; Xu et al., 2025; Sun et al., 2024; Tang et al., 2025). A natural intuition is that, within the capacity of a model, the longer the caption, the more fine-grained details it should include. However, our empirical study on Gemini-2.5 Pro's detailed captioning reveals a **"co-growth" phenomenon: as captions grow longer, not only does the proportion of correct fine-grained details increase, but the amount of hallucinated content also rises.**

As shown in Figure 2, the left blue y-axis denotes the detail ratio, which is the proportion of fine-grained information successfully captured. The right red y-axis depicts the hallucination ratio, the proportion of mentioned details that are factually incorrect despite being noticed by the model. The grey histogram shows the distribution of caption lengths produced under unconstrained generation. Both ratios rise together, indicating that in the unconstrained setting, richer descriptions are inherently accompanied by more hallucinated content.

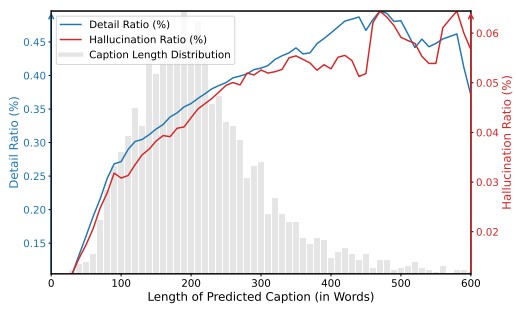

Figure 2: Relationship between caption length, detail coverage, and hallucination on Gemini-2.5-Pro in the detailed captioning task.

This "co-growth" of detail and hallucination exposes a fundamental challenge in fine-grained perception: short captions are safe but incomplete, missing subtle events, background cues, or cross-modal interactions. Overly long captions are richly descriptive but risk injecting content not grounded in the input, a critical flaw for applications requiring factual precision, such as assistive AI, scientific reporting, or autonomous agents. The difficulty is amplified in omni-modal settings, where models must jointly process asymmetric information densities across visual and auditory streams.

To address this, we design **Omni-Detective**, an agentic data pipeline in which an LLM agent plays the role of a "detective", invoking tool-calling (OCR, ASR, MLLM, etc.) and modality-specific observers to iteratively gather evidence. Each round incrementally adds valid, grounded details while cross-checking existing claims, explicitly targeting the decoupling of detail gain from hallucination growth. This yields detailed caption datasets with minimal noise.

Using data from the multi-round investigative pipeline, we train **Audio-Captioner** and **Omni-Captioner** with a two-stage curriculum: We first freeze the visual encoder to force precise alignment with sparse but critical audio cues. We then jointly optimize both modalities to produce coherent, cross-modal, and richly detailed narratives. This approach delivers strong empirical gains: Omni-Captioner sets a new state-of-the-art on VDC (Chai et al., 2025), and on the video-SALMONN 2 testset (Tang et al., 2025) achieves the most balanced trade-off between detail coverage and hallucination. In cascade caption-to-QA evaluations, Audio-Captioner attains the best results on MMAU (Sakshi et al., 2025) and MMAR (Ma et al., 2025), surpassing all open-source models and even outperforming Gemini-2.5-Flash, while Omni-Captioner achieves the highest overall score on Video-MME (Fu et al., 2025), Video-Holmes (Cheng et al., 2025), WorldSense (Hong et al., 2025), and Daily-Omni (Zhou et al., 2025) among existing open-source omni models. These results confirm that our training paradigm improves fine-grained perception across modalities.

To achieve a better evaluation for omni detailed perception, we introduce **Omni-Cloze**, the first cloze-style benchmark across audio-only, visual-only, and audio–visual scenarios, with high-quality human-verification. Omni-Cloze designs fine-grained details to form multiple-choice blanks, also including a "Not Given" option to explicitly distinguish omission from hallucination. The single-pass, automatic scoring protocol drastically reduces evaluation cost while ensuring stability and robustness. Our experiments and analysis demonstrate that the Omni-Detective pipeline and the Omni-Captioner model shift the detail–hallucination frontier outward: delivering richer descriptions without proportionally increasing hallucination, and achieving the best results on both existing detailed captioning benchmarks and the proposed Omni-Cloze. These results not only validate our design, but also highlight the importance of investigative data generation and evaluation paradigms tailored to the unique challenges of fine-grained omni-modal perception.

## 2 RELATED WORK

### 2.1 DETAILED PERCEPTION MODELS

Detailed perception, also referred to as detailed captioning, aims to generate fine-grained descriptions of an audio or video segment. This task differs from dense video captioning (Yang et al., 2023; Zhou et al., 2024; Geng et al., 2025), which requires models to output temporally-aligned captions across an entire long-form video. In contrast, detailed captioning focuses on producing descriptions that are as detailed as possible within short video or audio segments. The IIW Project (Garg et al., 2024) built a human-in-the-loop framework for curating hyper-detailed image descriptions, highlighting the role of expert-guided iterative refinement in generating high-quality annotations. In the video domain, AuroraCap (Chai et al., 2025) firstly explored the video detailed captioning task. More recently, video-SALMONN 2 (Tang et al., 2025) introduced a multi-round direct preference optimization (DPO) strategy to enhance models' capabilities in audio–visual detailed captioning and question-answering. However, most prior models are visual-centric, underutilizing the rich information present in the audio modality, including sound effects and events, speech content, and music cues, which can provide critical contextual details. Furthermore, the majority of these models rely on training data collected via manually designed prompting (Chai et al., 2025; Geng et al., 2025; Lu et al., 2025). This reliance leads to an inherent trade-off between precision and dataset scale of the descriptions, constraining the development of high-detail, low-hallucination multimodal perception models.

### 2.2 DETAILED PERCEPTION EVALUATION

Evaluating fine-grained captioning capability requires both (1) an effective evaluation metric and (2) a high-quality benchmark.

On the metric side, traditional measures such as BLEU (Papineni et al., 2002), METEOR (Banerjee & Lavie, 2005), and CIDEr (Vedantam et al., 2015), which were originally designed for machine translation or short-form captioning, struggle to faithfully assess captions containing long and information-rich descriptions. To address this, VDC (Chai et al., 2025) introduced an evaluation framework based on multiple short visual question–answer pairs derived from each prediction–reference pair. Specifically, for 1 detailed caption containing $k$ such QA pairs, VDC requires $2k$ LLM calls, raising concerns regarding both evaluation efficiency and the accumulation of evaluation error. Other approaches relying on counting events (Tang et al., 2025), or a cascade of captioning and QA evaluation (Tang et al., 2025; Lu et al., 2025), can not adequately evaluate the unique characteristics of the detailed captioning task. To overcome these limitations, Omni-Cloze adopts a cloze-style evaluation paradigm, which simultaneously addresses stability, efficiency, and reliability.

Table 1: Comparison of VDC and our proposed Omni-Cloze benchmark for detailed captioning. Omni-Cloze switches from multi-turn QA to a cloze-style paradigm, enabling a drastic reduction in LLM calls. Omni-Cloze also increases the number of evaluation questions, and expands modality coverage from purely visual to audio-only and audio–visual scenarios.

| Benchmark | Evaluation Method | Modality | | | Questions per Caption | LLM Calls per Caption |
| --- | --- | --- | --- | --- | --- | --- |
| | | Visual | Audio | AV | | |
| **VDC (Chai et al., 2025)** | Multiple QA | ✔ | ✗ | ✗ | 19 | 38 |
| **Omni-Cloze** *(Ours)* | Cloze | ✔ | ✔ | ✔ | **30** | **1** |

On the benchmark side, VDC focuses solely on the video modality. In contrast, Omni-Cloze covers audio-only, visual-only, and audio–visual scenarios, offering a more comprehensive and balanced evaluation of omni-modal detailed perception. Moreover, for each captioning instance, Omni-Cloze provides more questions. Table 1 illustrates the comparison between VDC and Omni-Cloze.

## 3 AGENTIC DATA GENERATION: OMNI-DETECTIVE

To enable high-quality fine-grained detailed captioning, we introduce **Omni-Detective**, an agentic data generation framework that leverages iterative *Query-Observation* cycles to autonomously extract and synthesize precise, richly detailed, and minimally hallucinatory audio–visual annotations. As illustrated in Figure 3, the core design philosophy draws on the analogy of a human detective: rather than making a single-pass observation, the detective strategically queries independent observers, leverages domain-specific tools, integrates multi-modal evidence, and incrementally refines its understanding before producing the final information.

Omni-Detective is composed of three key components: (1) a **Detective Agent** that spontaneously orchestrates the perception process, (2) a **Tool Box** containing multiple utilities (e.g., multimodal large language model (MLLM), optical character recognition (OCR), automatic speech recognition (ASR)) for extracting precise information from multimodal data, and (3) independent **Observers** that interact with raw video–audio streams to probe targeted aspects. The generation process unfolds in multiple steps: at each step, the Detective Agent formulates queries and invokes relevant tools; the Observers analyze the retrieved content and feed enriched observations back to the Agent; this iterative loop continues until sufficient fine-grained evidence is collected. Finally, the Agent integrates all observations into a highly detailed and minimally hallucinatory caption. This design allows adaptive allocation of perception effort, leveraging both structured tool-calls and free-form reasoning, thereby enabling scalable, accurate, and modality-complete detailed caption creation.

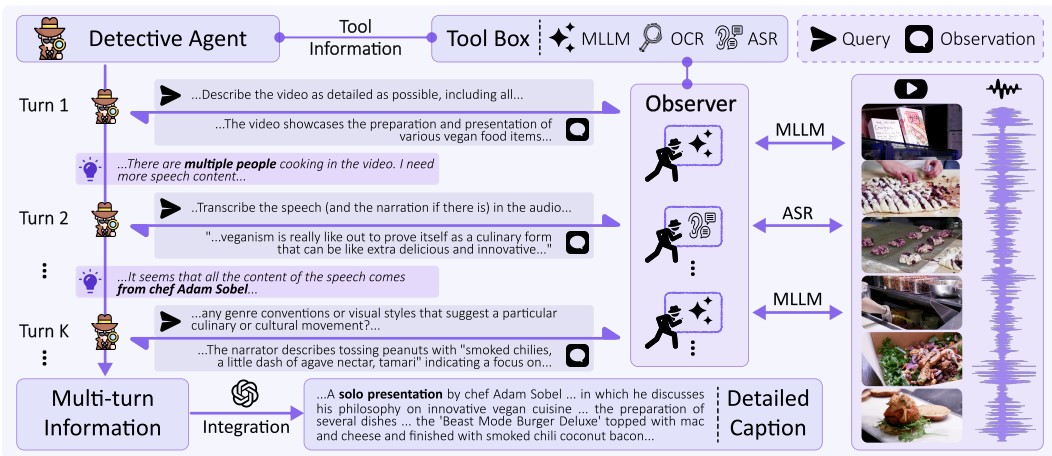

Figure 3: **Omni-Detective.** An agentic data generation and cleaning pipeline integrating specialist tools for omni detailed perception.

## 4 OMNI DETAILED CAPTIONING MODEL: OMNI-CAPTIONER

### 4.1 TRAINING STRATEGIES

Leveraging the high-fidelity multimodal detailed captioning data produced by Omni-Detective, we train **Audio-Captioner** and **Omni-Captioner** with a two-stage curriculum over the audio and audio–visual modalities. We adopt Qwen-2.5-Omni-7B (Xu et al., 2025) as the backbone, and empirically find that removing textual prompts in the input improves the model's descriptive performance; hence, both training stages are conducted without explicit text prompts. Comprehensive dataset statistics and hyperparameter settings are provided in Appendix A.

**Stage 1: Audio Perception Alignment** For audio–visual data segments of equal duration, the visual modality typically carries higher information density, which may lead the model to under-attend to sparse but semantically critical audio cues if trained jointly from the start. To mitigate this imbalance, we freeze the visual encoder and optimize only the audio encoder and the LLM using audio-only detailed captioning data. This stage enforces accurate grounding in the audio stream, resulting in the **Audio-Captioner**.

**Stage 2: Audio-Visual Perception Alignment** We then perform joint training on audio–visual detailed captioning data. In this stage, captions are notably longer, reaching an average of 1,125 words

per short video, reflecting the integration of both modalities. All model components are unfrozen for full-parameter fine-tuning, enabling the network to exploit cross-modal complementarities and produce rich, coherent, and modality-complete descriptions, resulting in the **Omni-Captioner**.

## 4.2 RESULTS ON EXISTING BENCHMARKS

To assess the effectiveness of **Omni-Detective** as a data pipeline, as well as the performance of the resulting **Audio-Captioner** and **Omni-Captioner** models, we conduct experiments under two evaluation settings:

1. **Direct evaluation** on existing detailed captioning benchmarks, which measure a model's ability to produce fine-grained and information-rich descriptions;

2. **Cascade evaluation** where the model first generates detailed captions, which are subsequently used to perform downstream question-answering tasks, thereby gauging the completeness of the produced captions.

It is worth noting that, to the best of our knowledge, no dedicated benchmark currently exists for detailed captioning in the audio-only modality. Consequently, under the first setting, we report results only for the Omni-Captioner model. Detailed settings are shown in Appendix C.1.

Table 2: Results on existing benchmarks for evaluating models' detailed captioning ability. The best-performing models in each category are highlighted in **bold**, and the second-best ones are underlined. All the results are obtained from their original papers.

| Model | Modality | VDC Detailed | | video-SALMONN $2_{test}$ | |
|---|---|---|---|---|---|
| | | Acc % ($\uparrow$) | Score ($\uparrow$) | Miss % ($\downarrow$) | Hall % ($\downarrow$) |
| *Proprietary Models* | | | | | |
| GPT-4o (GPT4oTeam, 2024) | V | 46.3 | 2.5 | 17.0 | 14.2 |
| Gemini 1.5 Pro (GeminiTeam, 2024) | A + V | 43.1 | 2.2 | 21.8 | 16.5 |
| *Open-Source Models* | | | | | |
| LLaVA-OneVision-7B (Li et al., 2024) | V | 41.2 | 2.1 | 23.3 | 27.4 |
| InternVideo2.5-7B (Wang et al., 2025) | V | 39.6 | 2.2 | 30.8 | 15.0 |
| Qwen2.5-VL-7B (Bai et al., 2025) | V | 44.5 | 2.4 | 21.9 | 17.4 |
| VideoLLaMA3-7B (Zhang et al., 2025) | V | 33.4 | 1.9 | 44.9 | 11.6 |
| VideoLLaMA2-7B (Cheng et al., 2024) | A + V | - | - | 56.8 | **8.9** |
| video-SALMONN-13B (Sun et al., 2024) | A + V | - | - | 52.1 | 26.6 |
| Qwen2.5-Omni-7B (Xu et al., 2025) | A + V | 39.7 | 2.2 | 26.3 | 21.7 |
| video-SALMONN2-7B (Tang et al., 2025) | A + V | 46.1 | 2.5 | **10.0** | 12.9 |
| **Omni-Captioner-7B** | A + V | **55.0** | **2.7** | 17.8 | 10.9 |

**Performance of Detailed Captioning.** The **VDC** benchmark (Chai et al., 2025) employs multiple short visual QA pairs to evaluate video-only detailed captioning performance. As shown in Table 2, Omni-Captioner achieves the highest accuracy (55.0%) and score (2.7), surpassing all proprietary and open-source baselines, thereby establishing a new state-of-the-art on VDC. The **video-SALMONN $2_{test}$** benchmark (Tang et al., 2025) measures the quality of omni-modal detailed captions by computing the missing rate (Miss%) and hallucination rate (Hall%) of events across both visual and auditory modalities. These two metrics are inherently a trade-off, as illustrated in Figure 2: increasing caption length typically reduces missing events but simultaneously raises hallucination rates. For example, VideoLLaMA2-7B produces shorter captions, resulting in the lowest hallucination (8.9%) but a high missing rate (56.8%), indicating poor coverage of fine-grained details. Conversely, video-SALMONN2-7B captures more details (10.0%) but at the cost of greater hallucination (12.9%). Despite being evaluated in a zero-shot setting (without knowing adaptation to the event distribution of video-SALMONN $2_{test}$), Omni-Captioner-7B achieves the best trade-off, attaining the second-lowest missing rate (17.8%) while keeping the second-lowest hallucination rate (10.9%), demonstrating its ability to deliver fine-grained yet reliable omni-modal descriptions.

Table 3: Results on existing benchmarks using a caption-to-QA cascade evaluation among strong audio-only and omni models.

(a) Audio models.

| Model | MMAU | MMAR |
|---|---|---|
| ***Proprietary Models*** | | |
| GPT-4o Audio | 62.4 | 59.3 |
| Gemini 2.0 Flash | 58.6 | 50.6 |
| Gemini 2.5 Flash | 65.6 | 58.2 |
| Gemini 2.5 Pro | 70.0 | 64.1 |
| ***Open-Source Models*** | | |
| SALMONN-13B | 58.36 | 42.5 |
| MiDashengLM-7B | 59.4 | 50.7 |
| Qwen2-Audio-7B | 63.3 | 44.2 |
| Qwen2.5-Omni-7B | 65.2 | 51.8 |
| **Audio-Captioner-7B** | **70.0** | **59.8** |

(b) Omni models.

| Model | Video -MME | Video -Holmes | World Sense | Daily -Omni |
|---|---|---|---|---|
| ***Proprietary Models*** | | | | |
| Gemini 2.0 Flash | 64.4 | 51.6 | 43.1 | 60.6 |
| Gemini 2.5 Flash | 69.1 | 52.8 | 44.6 | 59.5 |
| Gemini 2.5 Pro | 75.0 | 59.9 | 53.6 | 73.6 |
| ***Open-Source Models*** | | | | |
| video-SALMONN-13B | 41.8 | 31.4 | 26.8 | 45.0 |
| VideoLLaMA 2-7B | 44.4 | 33.5 | 26.7 | 39.9 |
| Qwen2.5-Omni-7B | 52.7 | 35.7 | 30.6 | 47.9 |
| video-SALMONN 2-7B | 65.9 | 42.9 | 44.1 | 59.7 |
| **Omni-Captioner-7B** | **67.1** | **48.8** | **48.2** | **67.9** |

**Performance of Applying Detailed Captioning.** An alternative perspective to assess detailed captioning quality is to adopt a caption-to-QA cascade: first generate audio or audio–visual captions, then feed them into a large language model to answer questions from existing QA benchmarks. Although cascaded inference may underperform end-to-end QA models on certain types of questions (e.g., precise counting), it serves as a faithful proxy for measuring the completeness of generated captions, thereby reflecting the quality of detailed perception. We evaluate **Audio-Captioner** on two audio-focused QA benchmarks: **MMAU** (Sakshi et al., 2025) and **MMAR** (Ma et al., 2025), and **Omni-Captioner** on four audio-video understanding and reasoning benchmarks: **Video-MME** (Fu et al., 2025), **Video-Holmes** (Cheng et al., 2025), **WorldSense** (Hong et al., 2025), and **Daily-Omni** (Zhou et al., 2025). In all cases, GPT-4o is used as the QA backbone to ensure consistent reasoning capability. As shown in Table 3a, for audio-only tasks, Audio-Captioner-7B achieves 70.0 on MMAU, matching the best proprietary model Gemini 2.5 Pro and outperforming all other open-source baselines by margins. On MMAR, which requires complex multi-step reasoning with mixed audio types, Audio-Captioner reaches 59.8, leading all open-source models and surpassing proprietary Gemini 2.0 Flash. These results indicate that the captions produced by Audio-Captioner preserve key audio details required for downstream QA task, even in acoustically cluttered mixed-modality scenarios in MMAR. For audio–visual tasks, Omni-Captioner-7B delivers the highest scores among open-source models across all four benchmarks, as shown in Table 3b. On Video-MME, which assesses comprehensive multimodal understanding, Omni-Captioner reaches 67.1, outperforming the previous state-of-the-art open-source detailed captioning model video-SALMONN 2. It also scores 48.8 on Video-Holmes, a high-order reasoning benchmark requiring multi-detail grounding. Similarly, it achieves 48.2 on WorldSense and 67.9 on Daily-Omni, which evaluate audio-visual synchronised and daily-life scenarios. Although proprietary models such as Gemini 2.5 Pro still lead in absolute terms, Omni-Captioner narrows the gap substantially while maintaining consistent gains over other open-source approaches. These trends suggest that high-quality, modality-rich captions from Omni-Captioner encode the fine-grained visual, audio, and cross-modal cues needed to support diverse and challenging downstream QA and reasoning tasks. We provide more extensive results and in-depth analysis in Appendix C.2.

## 5 OMNI DETAILED CAPTIONING BENCHMARK: OMNI-CLOZE

### 5.1 BENCHMARK OVERVIEW

Directly and reliably evaluating the detailed captioning ability of MLLMs is challenging due to the open-ended nature of the output space. To address this, we introduce **Omni-Cloze**, which frames evaluation as a **cloze-style multiple-choice proxy task**, as shown in Figure 4. Omni-Cloze is a unified benchmark for evaluating detailed captioning across **audio-only**, **visual-only**, and **audio–visual** settings. The dataset spans 9 main domains and 47 sub-categories, covering diverse topics such as

education, entertainment, sports, news, science, and lifestyle, with a total of 2k video clips with 70k fine-grained cloze blanks. This balanced, multi-domain, and modality-complete benchmark enables a comprehensive assessment of fine-grained multimodal perception. For full statistics, taxonomy design, and detailed distribution, please refer to Appendix B.2.

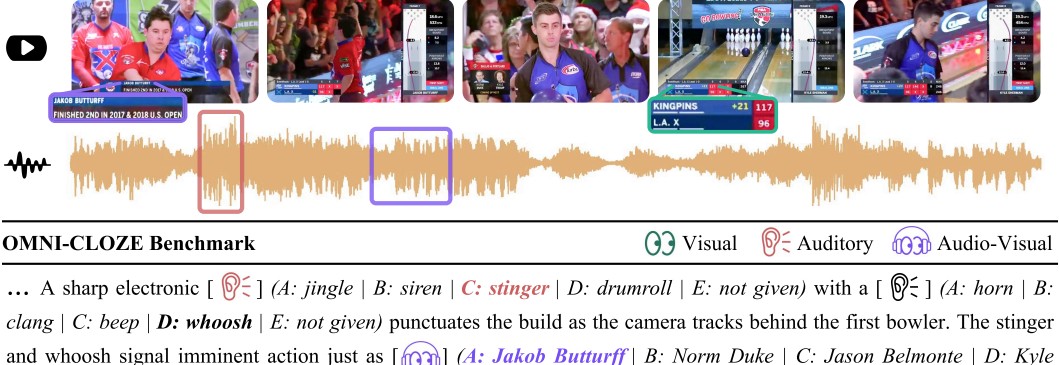

**OMNI-CLOZE Benchmark**      👀 Visual    🎧 Auditory    🎧 Audio-Visual

… A sharp electronic [ 🎧 ] *(A: jingle | B: siren | C: stinger | D: drumroll | E: not given)* with a [ 🎧 ] *(A: horn | B: clang | C: beep | D: whoosh | E: not given)* punctuates the build as the camera tracks behind the first bowler. The stinger and whoosh signal imminent action just as [ 🎧 ] *(A: Jakob Butturff | B: Norm Duke | C: Jason Belmonte | D: Kyle Sherman | E: not given)* begins his approach on screen. The left overlay shows "Semifinals - L.A. X Lead [ 👀 ] *(A: 0-1 | B: 1-0 | C: 1-1 | D: 2-0 | E: not given)*" with Kingpins [ 👀 ] *(A: 96 | B: 126 | C: 117 | D: 146 | E: not given)* …

Figure 4: **Omni-Cloze** utilizes cloze-style MCQ to evaluate models' detailed captioning abilities.

## 5.2 DATA CURATION

We design a multi-stage curation pipeline to construct **Omni-Cloze**, as shown in Figure 5. The process begins with collecting visual-only, audio-only, and audio–visual detailed captions from Omni-Detective. These captions are cross-validated across modalities to remove inconsistencies and hallucinations. Cloze-style passages are then generated by designing and masking semantically salient spans, as well as plausible but incorrect distractors are constructed for each blank. All cloze questions undergo human validation before inclusion in the final benchmark. A complete, step-by-step description of the data curation pipeline is provided in Appendix B.1.

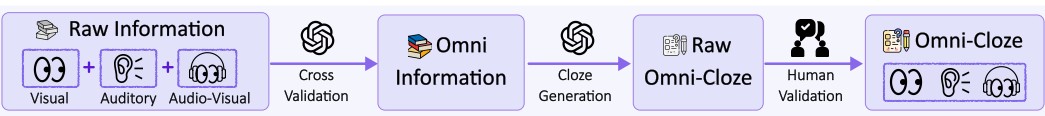

Figure 5: Data curation pipeline for Omni-Cloze.

## 5.3 EVALUATION PROTOCOL

For each evaluation sample, the model first generates a detailed caption, and an LLM is then tasked with filling in masked spans in the passage by selecting from several choices. To ensure stability and reliability, we introduce an additional *"Not Given"* option for each blank. We adopt *accuracy* as the main evaluation metric, and define a *hallucination* as a case where the model selects an incorrect option instead of choosing "Not Given". This design minimizes the subjective LLM reasoning, thereby improving evaluation stability. The design also enables a direct breakdown of model errors into either not-given rate or hallucination rate, ensuring reliable and interpretable measurement.

## 6 RESULTS AND ANALYSIS

### 6.1 RESULTS ON OMNI-CLOZE

Table 4 reports the results of both audio-only and omni models on the Omni-Cloze benchmark. For audio-only models (Table 4a), Audio-Captioner achieves a substantial lead with 53.2% accuracy, outperforming the strongest proprietary baseline of Gemini 2.5 Pro by 5.2% absolute points. Other open-source audio models remain below 30%, highlighting the difficulty of fine-grained audio perception. For omni models with audio and video abilities (Table 4b), the proposed Omni-Captioner

also achieves the best overall accuracy of 56.4%, surpassing all open-source and proprietary counterparts. It delivers strong performance across all modality splits, with 57.0% in visual-only, 54.5% in audio-only, and an outstanding 62.1% in the AV subset. These consistent improvements demonstrate that training with high-quality, modality-rich detailed captions enables significantly better fine-grained multimodal perception. We also report the not-given rate and hallucination rate of each model in Appendix C.3, providing further insights into the error characteristics of their predictions.

Table 4: Results on Omni-Cloze among strong audio-only models and omni models.

(a) Audio models.

| Model | Acc% ↑ |
|---|---|
| *Proprietary Models* | |
| GPT-4o Audio | 35.8 |
| Gemini 2.0 Flash | 20.0 |
| Gemini 2.5 Flash | 42.6 |
| Gemini 2.5 Pro | 48.0 |
| *Open-Source Models* | |
| SALMONN-13B | 10.6 |
| MiDashengLM-7B | 19.5 |
| Qwen2-Audio-7B | 22.2 |
| Qwen2.5-Omni-7B | 25.8 |
| **Audio-Captioner-7B** | **53.2** |

(b) Omni models.

| Model | Visual%↑ | Audio%↑ | AV%↑ | Total%↑ |
|---|---|---|---|---|
| *Proprietary Models* | | | | |
| Gemini 2.0 Flash | 32.3 | 31.7 | 40.1 | 33.2 |
| Gemini 2.5 Flash | 24.4 | 36.2 | 41.4 | 33.0 |
| Gemini 2.5 Pro | 40.8 | 44.1 | 52.8 | 43.6 |
| *Open-Source Models* | | | | |
| video-SALMONN-13B | 3.5 | 2.3 | 4.6 | 3.3 |
| VideoLLaMA 2-7B | 7.6 | 4.3 | 8.7 | 6.6 |
| Qwen2.5-Omni-7B | 18.3 | 14.1 | 21.9 | 16.6 |
| video-SALMONN 2-7B | 37.5 | 40.3 | 45.0 | 39.5 |
| **Omni-Captioner-7B** | **57.0** | **54.5** | **62.1** | **56.4** |

## 6.2 ANALYSIS OF OMNI-DETECTIVE

We conduct a fine-grained analysis of Omni-Detective behavior under different investigation budgets. Figure 6 reports the trends in detail rate, not-given rate, and hallucination rate as we increase the number of Omni-Detective steps applied to the same input. We observe that as the number of steps increases, the detail rate rises steadily, indicating that the iterative investigation process successfully extracts additional fine-grained information from the multimodal tools. Meanwhile, both the not-given rate and hallucination rate exhibit a downward trend: the reduction in not-given reflects improved coverage of relevant details, while the decline in hallucination suggests that the model can self-correct some prior incorrect inferences when provided with more investigative opportunities.

Interestingly, the hallucination rate converges relatively early (around step 5-6) in the Omni-Cloze benchmark, implying that current multimodal tools have a fundamental ceiling in eliminating incorrect claims: certain details are misclassified and are difficult to revise even with extended investigation and tool collaboration. In contrast, the detail rate continues to improve with more steps, showing that additional computation can still reveal new correct details. Such analysis highlights how Omni-Detective can probe the practical performance limits of existing multimodal perception data generation pipelines.

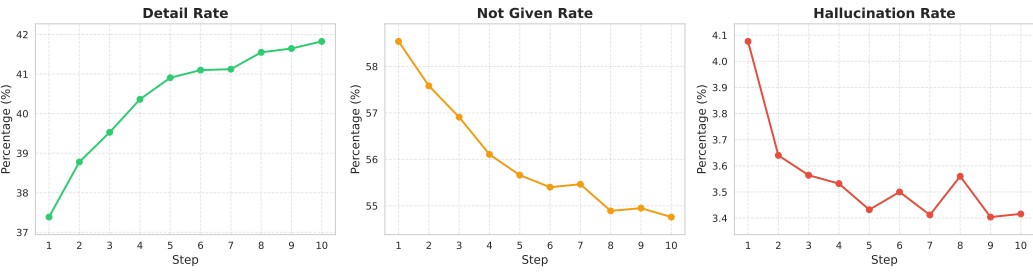

Figure 6: Trends of detail rate, not given rate, and hallucination rate vs steps in Omni-Detective.

We conduct a cascade evaluation on two representative benchmarks: MMAR for audio-only QA and Video-MME for audio–visual QA, with the same experimental setting described in Section 4.2. As shown in Table 5, applying the Omni-Detective pipeline (with Gemini 2.5 Pro as the MLLM) yields consistent improvements in downstream QA performance. We attribute

Table 5: Ablation on the effect of directly applying Omni-Detective to the caption-to-QA cascade.

| Method | MMAR | Video-MME |
|---|---|---|
| Gemini 2.5 Pro | 64.1 | 75.0 |
| + Omni-Detective | 68.3 | 76.1 |

these gains to Omni-Detective's ability to iteratively refine and expand captions, thereby increasing their completeness and coverage of fine-grained multimodal details. Richer and more accurate captions provide the QA backbone LLM with more relevant contextual information, enabling it to answer complex questions more reliably, especially in settings that require high detail recall across modalities.

## 6.3 ANALYSIS OF OMNI-CLOZE

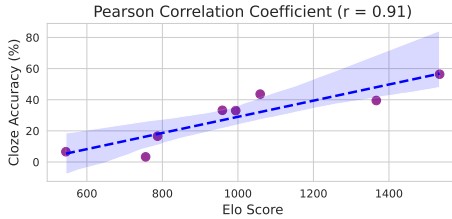

(a) Scatter plot between Elo scores and Omni-Cloze accuracy across models.

(b) Pearson correlation coefficients ($r$) between human-assigned Elo scores and benchmark-specific automatic evaluation metrics across models. Higher $r$ values indicate stronger agreement between human judgments and the benchmark metric.

| Benchmark | Metric | Pearson's $r$ |
|---|---|---|
| VDC | VDCscore | 0.86 |
| video-SALMONN $2_{test}$ | LLM-Judge | 0.83 |
| Omni-Cloze | Cloze-Acc | 0.91 |

Figure 7: Scatter plot across models and Pearson correlation coefficients across benchmark metrics.

We further examine the alignment between Omni-Cloze automatic evaluation and human preferences. Following an arena-style Elo evaluation (Chiang et al., 2024), we randomly sample outputs from two different models for the same audio–visual input, and ask annotators to rate the detailedness of their captions. Full details of the hyperparameters are provided in Appendix D.1.1.

Figure 7a visualizes the scatter plot between Elo scores and Omni-Cloze accuracy for individual models. We observe a strong positive correlation, indicating that higher scores on Omni-Cloze are consistently associated with higher human preference ratings. We further include a modality-wise analysis in Appendix D.1.2, where separate scatter plots are shown for the visual, audio, and audio–visual modalities. Table 7b reports Pearson correlation coefficients ($r$) between Elo scores and automatic metrics from three benchmarks. Omni-Cloze achieves the highest agreement of $r = 0.91$, exceeding both VDC and video-SALMONN $2_{test}$. All $r$ values are above 0.8, confirming that multi-turn QA, counting events, and cloze-style paradigms provide reliable measures of detailed captioning quality. We hypothesize that the superior correlation of Omni-Cloze arises from its evaluation design: the LLM is used only for information extraction in masked-span completion, with a single call per caption, thus avoiding multi-turn reasoning chains. This constrained usage reduces stochasticity and judgment drift, leading to a more stable evaluation and consistent agreement with human perception.

## 7 CONCLUSION

In this work, we presented a complete framework for advancing omni detailed perception from three perspectives: data pipeline, models, and benchmark. We proposed Omni-Detective, an agentic data generation pipeline that leverages tool-calling to collect highly detailed yet minimally hallucinatory captions. We leveraged these high-fidelity data to train Audio-Captioner and Omni-Captioner through a two-stage curriculum, achieving new state-of-the-art results and a superior detail–hallucination balance on diverse benchmarks. We introduced Omni-Cloze, the first cloze-style benchmark covering all three modality configurations, enabling stable, efficient, and reliable evaluation of fine-grained multimodal perception. We hope that this work will spur the community to develop more reliable and fine-grained multimodal perception systems, and to extend evaluation protocols that provide stable measurements and transparently reflect the actual abilities of model outputs.

## ACKNOWLEDGEMENT

This work was supported by the National Natural Science Foundation of China (No. U23B2018), Shanghai Municipal Science and Technology Major Project under Grant 2021SHZDZX0102, and Yangtze River Delta Science and Technology Innovation Community Joint Research Project (2024CSJGG1100). The work was also supported in part by the Research Grants Council of the Hong Kong Special Administrative Region, China, under Project CUHK 14202125 and Project CUHK 14200824.

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

APPENDIX TABLE OF CONTENTS

# A TRAINING DETAILS

## A.1 DATA DETAILS

### A.1.1 DATA CURATION DETAILS

Our Omni-Detective pipeline integrates multiple large multimodal models to collect, verify, and refine fine-grained multimodal captions. Specifically, we employ the Gemini series: Gemini 2.5 Pro (with thinking) and Gemini 2.5 Flash (without thinking), both capable of processing visual and auditory inputs; the GPT series: GPT-4o Audio for direct audio understanding and GPT-4o Transcribe for automatic speech recognition; as well as Qwen-2.5-Omni to improve data production efficiency through faster caption generation and multimodal grounding. We set the maximum number of Detective–Observer interaction steps to 10, allowing the agent to early stop when the investigative process converges. For the source data, we use VGGSound[2] as the audio-only dataset, covering diverse sound events and speech segments, and FineVideo[3] as the audio–visual dataset, containing rich multimodal events for detailed perception tasks. After quality filtering based on content diversity, we retain approximately 55k audio-only samples for Audio-Captioner and 15k audio–visual samples for Omni-Captioner.

### A.1.2 OMNI-DETECTIVE PROMPT TEMPLATE

---

**Detective Agent System Prompt (Audio-Only As An Example)**

---

You are a world-class audio detective. Your mission is to meticulously extract every last detail from a sound recording and generate a definitive, detailed description of this unknown sound clip.
**Focus on:**
1. **Signal-Level:** The physical characteristics of the recording itself (e.g., is it high or low fidelity, is there clipping, static, hiss, hum, or other artifacts? What is the estimated frequency range?).
2. **Perception-Level:** How the audio is perceived by a human ear (e.g., loudness, pitch, timbre, rhythm, clarity of speech, the spatial location of sounds).
3. **Semantic-Level:** The explicit meaning and events within the audio (e.g., the specific words spoken, identifiable sounds like a "door slamming" or a "dog barking," the melody and instrumentation of music).
4. **Cultural-Level:** The broader social and cultural context (e.g., the language and accent of speakers, the genre of music and its potential era/origin, the social setting implied by background sounds).
Please note that this breakdown into levels is merely meant to provide you with some areas of focus. Your final description should not adhere to this structure, nor is it limited to these points.
**Your Method:**
You cannot listen to the audio directly. Instead, you have access to several independent observers who have listened to it. You can interrogate them to gather clues.
- You have a total of {max_calls} inquiries.
- You have specific analysis tools from {tool_list}.
- For each turn, you must select **one tool** and ask **one question**, either general or specific. Question examples: "What language is being spoken?", "Describe the background music.", "Did you hear any sounds other than speech?", "What was the emotional tone of the woman's voice?".
- **Crucially, the observers are not perfect.** Their reports may be incomplete, biased, or contain errors. You must act as a detective: cross-reference their answers, identify contradictions, and build a case for what is actually in the audio.
**Output format each turn (JSON):**

```
{
  "tool": "<tool_name_from_tool_list>",
  "question": "<your_question_text>"
}
```

**Your Final Output:**
When you have used all {max_calls} inquiries (i.e., when 0 remain), you must stop asking questions and produce your final description.
Your final output must be exceedingly detailed. Provide a chronological storyline of every noticeable element within the audio, including but not limited to a full speech-to-text transcription, and all signif-

---

[2]https://github.com/hche11/VGGSound
[3]https://huggingface.co/datasets/HuggingFaceFV/finevideo

icant audio events. This storyline, however, is merely a foundation; your comprehensive description must extend far beyond this basic information.
Do not output any additional information.

---

**Detective Agent Prompt**
──────────────────────────────────────────────────────────────

Observation: {observation}
Inquiries remaining: {calls_left}
**Tool and Question:**

---

**Observer System Prompt (LALM As An Example)**
──────────────────────────────────────────────────────────────

You are a hyper-perceptive audio analysis system. Your purpose is to act as a perfect, all-knowing sensor for an audio detective. You have been given an audio file.
**Your Core Mandate:**
You must provide **accurate, detailed, and objective answers** based *only* on the provided audio. You perceive everything within the audio with perfect clarity across all analytical levels.
**Strict Constraints:**
- **Be 100% Factual:** Your answers must be completely true to the audio. There is no room for error, uncertainty, or personal bias.
- **Maintain Your Persona:** You are a data source, an analytical tool. Never reveal that you are an AI or that you are working from a prompt. Respond directly and impersonally.
- **Be a Passive Tool:** You exist only to answer. Do not ask questions back, or try to guide the detective.
- **Be as detailed as possible:** The detective has a limited number of questions, so your responses must include all relevant details and nuances.

---

**Observer Prompt**
──────────────────────────────────────────────────────────────

Question: {question}

## A.2 MODEL DETAILS

Table 6 lists the training hyperparameters for Audio-Captioner and Omni-Captioner.

Table 6: Hyperparameters for training Audio-Captioner and Omni-Captioner.

| Model | Audio-Captioner | Omni-Captioner |
|---|---|---|
| GPU Usage | $8 \times$ A100 80GB | |
| Batch Size per GPU | 2 | 1 |
| Gradient Accumulation | 4 | 2 |
| Training Time | 8 Hours | 38 Hours |
| Training Epochs | 2 | |
| Peak Learning Rate | 5e-6 | |
| Learning Rate Scheduler | Linear | |
| Optimizer | AdamW | |

## B BENCHMARK DETAILS

### B.1 DATA CURATION DETAILS

**Step 1: Raw information acquisition.** We use the raw videos from the FineVideo dataset as the source material. From each full-length video, we extract a semantically complete segment of **10–60** seconds in duration, ensuring coherent context and preserving the integrity of events. For each selected segment, we collect detailed captions in three distinct modality configurations: visual-only,

audio-only, and audio–visual, sourced from the Omni-Detective data generation pipeline. Each caption contains rich, atomic details capturing visual objects and actions, auditory events, and cross-modal interactions.

**Step 2: Cross-modal validation.** To remove inconsistencies and hallucinations, captions undergo an omni-information verification step, where multiple modality-specific observations are cross-checked using LLM-based reasoning. This process consolidates reliable details from all modalities into a modality-complete omni-information description.

**Step 3: Human-in-the-loop Validation.** To ensure the absolute factual accuracy of the benchmark's foundation, we ask professional annotators to review the omni-information descriptions against the original video clips. Any remaining hallucinations, factual inaccuracies, or temporal misalignments are manually corrected. Only these gold standard descriptions, verified for their modality faithfulness and semantic completeness, serve as the source text for subsequent question generation.

**Step 3: Cloze Generation.** The validated descriptions are then converted into a cloze-format evaluation set via an automated procedure. We prompt an LLM to identify and mask semantically salient spans, such as specific entities, actions, attributes, and temporal markers, to form fill-in-the-blank questions. For each passage, **30** cloze items are generated with a strict modality-balanced distribution: at least **8** questions grounded in visual details, **8** in audio details, and **4** requiring audio–visual cross-modal grounding. The remaining blanks are allocated in an information-adaptive manner based on the density of available details. This approach ensures a fine-grained, high-quality dataset while enabling efficient and objective downstream evaluation.

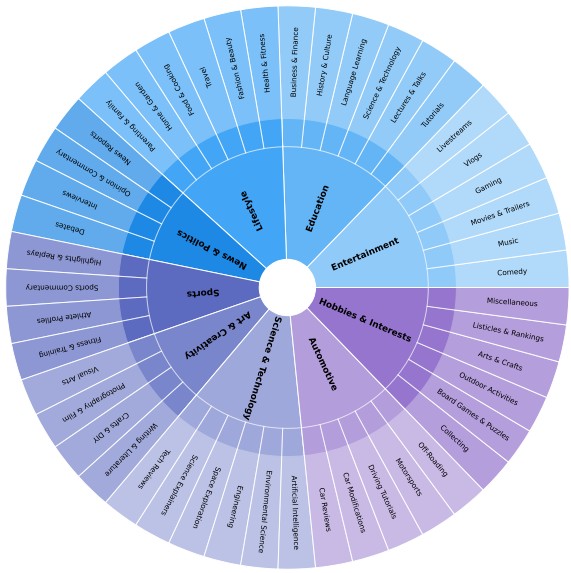

(b) Overall Statistics

| Statistics | Number |
|---|---|
| Total Videos | 2320 |
| Total Details | 69600 |
| Domain Categories | 9 |
| Sub-Categories | 47 |
| Avg. Passage Length | 444.6 words |
| Avg. Video Length | 34.2 sec |

(c) Modality-Wise Statistics

| Statistics | Audio | Visual | AV |
|---|---|---|---|
| Total Details | 23413 | 36362 | 9825 |
| ratio (%) | 33.6 | 52.2 | 14.1 |
| Avg. Details per Passage | 10.1 | 15.7 | 4.2 |

(a) Domain Distribution

Figure 8: **(a)** The domain distribution of Omni-Cloze. **(b)** Overall statistics of Omni-Cloze. **(c)** Modality-wise statistics across audio, visual, and audio-visual of Omni-Cloze.

## B.2 BENCHMARK STATISTICS

Omni-Cloze provides a unified testbed that jointly covers detailed perception in all three modality configurations, enabling fine-grained and balanced assessment. With a taxonomy design similar to FineVideo, as illustrated in Figure 8a, Omni-Cloze spans **9** main categories and **47** sub-categories, including diverse sources such as Education, Entertainment, Sports, News & Politics, Science & Technology, Automotive, Art & Creativity, Hobbies & Interests, and Lifestyle. We curated the dataset to achieve a balanced distribution across these categories, ensuring that each domain is sufficiently represented for comprehensive performance analysis. The benchmark contains **2,320** video clips and **69,600** annotated cloze blanks, each paired with long-form, detail-rich captions averaging

**444.6** words, over an average clip length of **34.2** seconds, as shown in Table 8b. On average, an audio-only passage contains **11.5** audio-only details, **13.0** video-only details, and **5.5** audio-visual details, as shown in Table 8c. Overall, Omni-Cloze offers a large-scale, multi-domain, and modality-complete benchmark, specifically designed for a effective evaluation of fine-grained multimodal perception.

## B.3 OMNI-CLOZE PROMPT TEMPLATE

---

**Cloze Generation Prompt**

───────────────────────────────────────────────────────────────────────────

You are given descriptions of the same recording:
1. **Audio description** — focuses only on what can be heard in the recording.
2. **Video description** — focuses only on what can be seen in the recording.
3. **Audio-video description** — includes both what is seen and what is heard.
**Your task:**
Create a **{total_number}-blank cloze test** from the given descriptions. The goal is to test a model's **detailed audio-video captioning and comprehension ability**. Each blank must target a **specific, concrete detail** from either:
- The audio track (speech, sound effects, music, background noises, tone, pacing, etc.), OR
- The video track (objects, people, actions, positions, colors, facial expressions, gestures, environment details, etc.), OR
- A combination of both audio and video features (synchronized events, actions matched with sounds, timing relationships, etc.)
**Important constraints:**
- Do **not** create blanks that require cultural or background knowledge beyond what is directly perceivable in the clip.
- Time-related aspects are acceptable in a general sense, but do **not** create blanks for specific timestamps or exact durations.
- Every question must be answerable purely from the described scene's observable and audible details.
**Instructions for creating the cloze test:**
1. Convert the given descriptions into a unified sequence of sentences to form the cloze test passage.
2. Select {total_number} important factual points to remove as blanks ([BLANK_n]).
3. Each blank must clearly relate to a perceptible fact from:
- "audio" — Determinable only from audio cues. **At least {audio_number} blanks**.
- "visual" — Determinable only from video cues. **At least {visual_number} blanks**.
- "audio-visual" — Requires combining both audio & visual information. **At least {av_number} blanks**.
4. For each blank, provide:
- The correct answer
- 3 **plausible but incorrect** distractor options, which:
* Are close in type or category to the correct answer
* Could plausibly be guessed by someone with incomplete or shallow understanding of the clip
* Are believable but **not** trivially absurd
- Label each blank with "required_modality":
* "audio" — Determinable only from audio content.
* "visual" — Determinable only from visual content.
* "audio-visual" — Requires observation of both audio & visual details.
5. Number the blanks clearly from 1 to {total_number}.
6. Avoid making blanks guessable from generic context only.
**Output format (JSON):**

```
{
  "passage": "A single string containing the cloze test text with 30
    placeholders like [BLANK_1], [BLANK_2], ... embedded at the exact
    locations in the passage.",
  "blanks": [
    {
      "number": 1,
      "answer": "string (the correct answer for BLANK_1)",
      "distractors":
        ["string (plausible wrong option 1)",
        "string (plausible wrong option 2)",
        "string (plausible wrong option 3)"],
```

---

```
      "required_modality":
        "string (one of \"audio\", \"visual\", \"audio-visual\")"
    },
    {
      "number": 2,
      "answer": "string (the correct answer for BLANK_2)",
      "distractors":
        ["wrong option 1",
        "wrong option 2",
        "wrong option 3"],
      "required_modality":
        "string (one of \"audio\", \"visual\", \"audio-visual\")"
    }
    // ... repeat until number \{total\_number\}
  ]
}
```
**Your Input:**
- Audio Description:
{audio_description}
- Video Description:
{video_description}
- Audio-Video Description:
{audio_video_description}
**Step summary for the model:**
1. Merge descriptions into a single gold standard text.
2. Create passage with {total_number} blanks.
3. Make sure:
- at least {audio_number} blanks have "required_modality": "audio".
- at least {visual_number} blanks have "required_modality": "visual".
- at least {av_number} blanks have "required_modality": "audio-visual".
4. Output strictly in the JSON format shown above.

---

**Cloze Completion Prompt**
________________________________________________________________________________________

You will be given two inputs:
1. A cloze (fill-in-the-blank) task, consisting of a passage describing a scene, with {number} blanks.
Each blank has 5 possible choices (A-D are specific contents, E means "not given").
2. A caption ({modality} description of the scene).
**Your task:**
- Carefully read the caption and use it to determine the correct answer for each blank.
- If the caption mentions the information corresponding to a blank, choose the matching option (A-D).
- If the caption does NOT mention that detail and it cannot be reasonably inferred from the given information, choose E ("not given").
- Only use general/common knowledge reasoning if it is strongly justified (for example, hearing a V10 engine can suggest a roadster).
- Unless the information is fully certain, do **NOT** guess.
- For each blank, output the letter (A/B/C/D/E) along with the answer.
- Output the final answer as a JSON object where keys are the blank numbers (as strings) and values are the option letters with the answer.
- Do **NOT** include any explanation or extra text in your output.
**Output format example:**

```
{
    "1": "A: xxx",
    "2": "E: xxx",
    "3": "D: xxx"
    // ... until \{number\}
}
```

Now process the following input:
**Cloze passage:**
{cloze}
**Caption:**

> {prediction}
> **Output:**

## C  MORE RESULTS

### C.1  EVALUATION DETAILS

For all evaluations, system prompts are configured to follow each model's recommended best-practice settings; if no official prompt is provided, we use the default instruction "You are a helpful assistant". Open-source models are decoded using greedy search (beam size = 1) without sampling for reproducibility. For proprietary models, Gemini 2.5 Pro is evaluated with its default thinking mode enabled, whereas other proprietary models are evaluated without thinking. In the case of cascade-based evaluations, we experimented with multiple prompt formulations for each model and adopted the best ones.

### C.2  RESULTS WITH CASCADE EVALUATION

#### C.2.1  AUDIO QA BENCHMARKS.

**MMAU** (Sakshi et al., 2025) benchmark evaluates both audio understanding and simple reasoning across sound, music, and speech domains, at three difficulty levels including easy, medium, and hard. As shown in Table 7, Audio-Captioner-7B achieves the highest average score of 70.0% among all models, matching the top proprietary model Gemini 2.5 Pro and surpassing all other open-source models. This significant improvement on easy tasks indicates that the model accurately captures key content in audio streams, producing captions that preserve all major information required for downstream understanding and reasoning.

**MMAR** (Ma et al., 2025) benchmark measures complex, multi-step audio reasoning, including scenarios mixing sound, music, and speech in various combinations. From Table 8, Audio-Captioner attains the best open-source performance with an average of 59.8%, and even surpasses the proprietary Gemini 2.5 Flash model. Notably, the performance gains of Audio-Captioner are even more pronounced in mixed-modality scenarios: in the challenging mixed Speech–Sound–Music scenario, Audio-Captioner achieves 66.67%, outperforming the next-best open-source model of Qwen-2.5-Omni by +37.5 absolute points. This underscores its robustness in fine-grained comprehension, especially in acoustically cluttered and compositionally complex environments.

#### C.2.2  AUDIO–VISUAL QA BENCHMARKS.

**Video-MME** (Fu et al., 2025) evaluates comprehension of videos spanning short (<2min), medium (4–15min), and long (30–60min) durations. As shown in Table 9, Omni-Captioner achieves an average of 67.1% without subtitles, ranking first among open-source models and closely trailing Gemini 2.5 Flash. It delivers prominent short-video performance of 77.2%, but lags on long videos of 58.0%, suggesting a potential avenue for future improvement.

**Video-Holmes** (Cheng et al., 2025) measures higher-order video reasoning across seven sub-tasks requiring multi-detail grounding (e.g., Social Reasoning, Temporal Causal Inference). From Table 10, Omni-Captioner achieves an average score of 48.8%, outperforming the next-best open-source model, video-SALMONN 2 by +5.9 absolute points, with consistent gains across nearly all sub-task categories.

**Daily-Omni** (Zhou et al., 2025) benchmark evaluates temporally-aligned multimodal reasoning over diverse scenarios. As shown in Table 11, Omni-Captioner reaches the highest open-source score of 67.9%, and notably surpasses Gemini 2.5 Flash. It particularly excels in the AV Event Alignment task, where its score of 61.3% surpasses the next-best open-source model by +11.3, which highlights Omni-Captioner's strong capability to produce captions that reflect the temporal alignment of audio and visual details.

**WorldSense** (Hong et al., 2025) benchmark tests recognition, understanding, and reasoning in audio–visual perception in Table 12. Omni-Captioner achieves 48.2%, leading all open-source com-

petitors and closing much of the gap to proprietary Gemini 2.5 Pro, with balanced gains across all three skill dimensions.

## C.3    RESULTS WITH OMNI-CLOZE

Tables 13 and 14 report the *Accuracy* (Acc), *Not-Given rate* (NG), and *Hallucination rate* (Hall) for models with strong captioning ability on the Omni-Cloze benchmark, split into audio-only and audio–visual settings. Across both audio-only and audio–visual evaluations, our captioner models consistently produce lower Not-Given rates and higher accuracies than all open-source baselines, and often outperform strong proprietary systems. The increase in hallucination is a natural consequence of reduced abstention and richer detail coverage, also shown in Figure 2. This trade-off, given the magnitude of the accuracy gains, is favorable for fine-grained multimodal perception tasks. These trends confirm that the high-quality, modality-rich captions produced by Audio-/Omni-Captioner effectively support stable, detail-oriented cloze-style evaluation while revealing each model's strengths and weaknesses in coverage versus correctness.

Table 7: MMAU results with a cascade of audio captions and LLM. Results of three domains: Sound (**So**), Music (**Mu**), and Speech (**Sp**), and three difficulty levels **Easy**, **Medium**, and **Hard** are listed.

| Methods | Across Modality (%) | | | Across Difficulty (%) | | | Avg (%) |
|---|---|---|---|---|---|---|---|
| | So | Mu | Sp | Easy | Medium | Hard | |
| *Proprietary Models* | | | | | | | |
| GPT-4o Audio | 60.96 | 58.08 | 68.17 | 53.12 | 68.52 | 57.20 | 62.40 |
| Gemini 2.0 Flash | 60.66 | 57.49 | 57.66 | 45.98 | 65.19 | 55.51 | 58.60 |
| Gemini 2.5 Flash | 65.47 | 59.58 | 71.77 | 57.59 | 68.70 | 66.10 | 65.60 |
| Gemini 2.5 Pro | 68.47 | 65.87 | 75.68 | 66.07 | 73.70 | 65.25 | 70.00 |
| *Open-Source Models* | | | | | | | |
| SALMONN-13B | 67.57 | 60.36 | 47.15 | 49.11 | 64.38 | 53.39 | 58.36 |
| Qwen2-Audio-7B-Instruct | **71.47** | 58.98 | 47.75 | 46.43 | 65.37 | 58.05 | 59.40 |
| MiDashengLM-7B | **71.47** | 62.28 | 56.16 | 57.14 | 68.10 | 58.05 | 63.30 |
| Qwen2.5-Omni-7B | 69.67 | 64.67 | 61.26 | 58.48 | 69.07 | 62.71 | 65.20 |
| **Audio-Captioner-7B** | 67.87 | **66.17** | **75.98** | **70.09** | **71.85** | **65.68** | **70.00** |

Table 8: MMAR results with a cascade of audio captions and LLM. Results on single-modality: (**So**, **Mu**, **Sp**), and mixed-modality conditions: (**So-Mu**, **So-Sp**, **Mu-Sp**, **So-Mu-Sp**) are listed.

| Model | Single Modality (%) | | | Mixed Modalities (%) | | | | Avg (%) |
|---|---|---|---|---|---|---|---|---|
| | So | Mu | Sp | So-Mu | So-Sp | Mu-SP | So-Mu-Sp | |
| *Proprietary Models* | | | | | | | | |
| GPT-4o Audio | 41.21 | 41.75 | 69.05 | 36.36 | 73.39 | 69.51 | 62.50 | 59.30 |
| Gemini 2.0 Flash | 40.00 | 47.09 | 57.14 | 27.27 | 57.34 | 48.78 | 29.17 | 50.60 |
| Gemini 2.5 Flash | 42.42 | 44.17 | 68.03 | 63.64 | 67.43 | 64.63 | 58.33 | 58.20 |
| Gemini 2.5 Pro | 52.12 | 47.57 | 74.15 | 63.64 | 74.31 | 68.29 | 58.33 | 64.10 |
| *Open-Source Models* | | | | | | | | |
| SALMONN-13B | 50.91 | 35.92 | 43.20 | 9.09 | 46.79 | 37.80 | 25.00 | 42.50 |
| Qwen2-Audio-7B-Instruct | 46.06 | 40.29 | 60.88 | 27.27 | 53.67 | 46.34 | 45.83 | 50.70 |
| MiDashengLM-7B | 47.88 | 43.20 | 42.86 | **63.64** | 45.87 | 45.12 | 16.67 | 44.20 |
| Qwen2.5-Omni-7B | 43.64 | 43.69 | 64.29 | 27.27 | 51.83 | 53.66 | 29.17 | 51.80 |
| **Audio-Captioner-7B** | **55.15** | **46.12** | **68.37** | **63.64** | **61.01** | **67.07** | **66.67** | **59.80** |

Table 9: Video-MME results with a cascade of audio-visual captions and LLM. Results on **Short**, **Medium**, and **Long** durations are listed. *Results of Qwen2.5-Omni-7B and video-SALMONN 2-7B are from the cascade evaluation in the video-SALMONN 2 original paper.

| Model | Duration(%) | | | Avg(%) |
| --- | --- | --- | --- | --- |
| | Short | Medium | Long | |
| *Proprietary Models* | | | | |
| Gemini 2.0 Flash | 71.2 | 64.1 | 57.9 | 64.4 |
| Gemini 2.5 Flash | 76.2 | 70.2 | 60.8 | 69.1 |
| Gemini 2.5 Pro | 80.8 | 77.1 | 67.1 | 75.0 |
| *Open-Source Models* | | | | |
| video-SALMONN-13B | 42.8 | 39.4 | 43.1 | 41.8 |
| VideoLLaMA 2-7B | 43.7 | 44.9 | 44.7 | 44.4 |
| Qwen2.5-Omni-7B* | - | - | - | 52.7 |
| video-SALMONN 2-7B* | - | - | - | 65.9 |
| **Omni-Captioner-7B** | **77.2** | **66.1** | **58.0** | **67.1** |

Table 10: Video-Holmes results with a cascade of audio-visual captions and LLM. The benchmark covers seven sub-tasks: Social Reasoning (**SR**), Physical Anomaly Reasoning (**PAR**), Multimodal Hint Reasoning (**MHR**), Intention & Motive Chaining (**IMC**), Timeline Analysis (**TA**), Temporal Causal Inference (**TCI**), and Core Theme Inference (**CTI**).

| Model | Task(%) | | | | | | | Avg(%) |
| --- | --- | --- | --- | --- | --- | --- | --- | --- |
| | SR | IMC | CTI | TCI | TA | MHR | PAR | |
| *Proprietary Models* | | | | | | | | |
| GPT-4o | 50.0 | 43.5 | 39.6 | 39.6 | 35.5 | 45.8 | 41.2 | 42.7 |
| Gemini 2.0 Flash | 61.3 | 60.9 | 49.2 | 42.9 | 41.5 | 51.2 | 50.0 | 51.6 |
| Gemini 2.5 Flash | 59.9 | 59.0 | 46.7 | 46.5 | 43.5 | 54.8 | 56.7 | 52.8 |
| Gemini 2.5 Pro | 68.8 | 65.9 | 51.8 | 58.2 | 49.0 | 63.6 | 56.2 | 59.9 |
| *Open-Source Models* | | | | | | | | |
| video-SALMONN-13B | 35.6 | 35.9 | 23.3 | 26.0 | 36.5 | 28.6 | 36.6 | 31.4 |
| VideoLLaMA 2-7B | 35.3 | 38.8 | 27.0 | 27.8 | 34.0 | 34.9 | 37.1 | 33.5 |
| Qwen2.5-Omni-7B | 37.3 | 38.0 | 30.0 | 30.0 | 38.0 | 38.3 | 39.2 | 35.7 |
| video-SALMONN 2-7B | 53.1 | 42.8 | 40.0 | 39.2 | 37.0 | 41.0 | 46.4 | 42.9 |
| **Omni-Captioner-7B** | **57.2** | **55.8** | **40.4** | **45.8** | **40.5** | **48.5** | **51.5** | **48.8** |

Table 11: Daily-Omni results with a cascade of audio-visual captions and LLM. Daily-Omni spans diverse daily life scenarios.

| Model | Context Understanding | Reaso-ning | Infer-ence | Compa-rative | Event Sequence | AV Event Alignment | Avg |
| --- | --- | --- | --- | --- | --- | --- | --- |
| *Proprietary Models* | | | | | | | |
| Gemini 2.0 Flash | 51.3 | 74.3 | 76.0 | 69.5 | 54.9 | 50.4 | 60.6 |
| Gemini 2.5 Flash | 50.8 | 71.4 | 77.9 | 60.3 | 55.2 | 50.8 | 59.5 |
| Gemini 2.5 Pro | 66.3 | 87.4 | 83.8 | 80.9 | 65.7 | 68.9 | 73.6 |
| *Open-Source Models* | | | | | | | |
| video-SALMONN-13B | 37.3 | 58.9 | 56.5 | 51.9 | 39.9 | 36.6 | 45.0 |
| VideoLLaMA 2-7B | 31.6 | 49.7 | 51.9 | 48.1 | 37.3 | 30.7 | 39.9 |
| Qwen2.5-Omni-7B | 36.8 | 68.6 | 60.4 | 54.2 | 40.2 | 39.9 | 47.9 |
| video-SALMONN 2-7B | 51.3 | 74.9 | 74.7 | 65.6 | 53.9 | 50.0 | 59.7 |
| **Omni-Captioner-7B** | **61.7** | **80.0** | **83.1** | **69.5** | **61.8** | **61.3** | **67.9** |

Table 12: Worldsense results with a cascade of audio-visual captions and LLM. The benchmark tests **Recognition**, **Understanding**, and **Reasoning** covering broad scenarios.

| Model | Task Domain(%) | | | Avg(%) |
|---|---|---|---|---|
| | **Recognition** | **Understanding** | **Reasoning** | |
| *Proprietary Models* | | | | |
| Gemini 2.0 Flash | 38.0 | 45.8 | 49.2 | 43.1 |
| Gemini 2.5 Flash | 39.6 | 47.4 | 50.4 | 44.6 |
| Gemini 2.5 Pro | 48.8 | 57.3 | 57.9 | 53.6 |
| *Open-Source Models* | | | | |
| video-SALMONN-13B | 24.0 | 27.3 | 31.1 | 26.8 |
| VideoLLaMA 2-7B | 24.0 | 28.0 | 29.8 | 26.7 |
| Qwen2.5-Omni-7B | 27.2 | 34.1 | 32.7 | 30.6 |
| video-SALMONN 2-7B | 40.2 | 45.4 | 49.3 | 44.1 |
| **Omni-Captioner-7B** | **43.4** | **50.0** | **54.6** | **48.2** |

Table 13: Results for audio models with strong captioning ability on Omni-Cloze (Audio Part).

| Model | NG(↓) | Hall(↓) | **Acc(↑)** |
|---|---|---|---|
| *Proprietary Models* | | | |
| GPT-4o Audio | 60.27 | 3.96 | 35.77 |
| Gemini 2.0 Flash | 77.14 | 2.91 | 19.95 |
| Gemini 2.5 Flash | 51.22 | 6.19 | 42.59 |
| Gemini 2.5 Pro | 47.30 | 4.74 | 47.96 |
| *Open-Source Models* | | | |
| SALMONN-13B | 87.36 | 2.09 | 10.55 |
| MiDashengLM-7B | 77.50 | 3.00 | 19.50 |
| Qwen2-Audio-7B | 74.87 | 2.98 | 22.15 |
| Qwen2.5-Omni-7B | 71.41 | 2.78 | 25.80 |
| **Audio-Captioner-7B** | 40.99 | 5.81 | 53.19 |

Table 14: Results for audio-visual models with strong captioning ability on Omni-Cloze.

| Model | Visual Part% | | Acc(↑) | Audio Part% | | Acc(↑) | AV Part% | | Acc(↑) | Total% | | Acc(↑) |
|---|---|---|---|---|---|---|---|---|---|---|---|---|
| | NG(↓) | Hall(↓) | | NG(↓) | Hall(↓) | | NG(↓) | Hall(↓) | | NG(↓) | Hall(↓) | |
| *Proprietary Models* | | | | | | | | | | | | |
| Gemini 2.0 Flash | 62.3 | 5.4 | 32.3 | 65.4 | 3.0 | 31.7 | 54.4 | 5.5 | 40.1 | 62.2 | 4.6 | 33.2 |
| Gemini 2.5 Flash | 72.8 | 2.8 | 24.4 | 57.8 | 6.0 | 36.2 | 52.7 | 5.8 | 41.4 | 62.1 | 4.9 | 33.0 |
| Gemini 2.5 Pro | 53.3 | 5.9 | 40.8 | 52.2 | 3.7 | 44.1 | 40.7 | 6.5 | 52.8 | 51.1 | 5.2 | 43.6 |
| *Open-Source Models* | | | | | | | | | | | | |
| video-SALMONN-13B | 95.0 | 1.5 | 3.5 | 97.0 | 0.7 | 2.3 | 93.7 | 1.7 | 4.6 | 95.5 | 1.3 | 3.3 |
| VideoLLaMA 2-7B | 90.8 | 1.7 | 7.6 | 94.9 | 0.8 | 4.3 | 89.8 | 1.6 | 8.7 | 92.0 | 1.4 | 6.6 |
| Qwen2.5-Omni-7B | 80.0 | 1.8 | 18.3 | 83.2 | 2.7 | 14.1 | 74.9 | 3.1 | 21.9 | 80.9 | 2.5 | 16.6 |
| video-SALMONN 2-7B | 55.5 | 6.9 | 37.5 | 55.6 | 4.2 | 40.3 | 48.1 | 6.9 | 45.0 | 54.5 | 6.0 | 39.5 |
| **Omni-Captioner-7B** | 33.4 | 12.0 | 54.6 | 35.6 | 7.4 | 57.0 | 26.5 | 11.5 | 62.1 | 33.2 | 10.4 | 56.4 |

## D  MORE ANALYSIS

### D.1  ELO OF OMNI-CLOZE

#### D.1.1  HYPERPARAMETERS

To examine the correlation between the automatic evaluation results of Omni-Cloze and human preferences, we conduct a human-judged Elo ranking across a set of audio–video models, including Gemini 2.0 Flash, Gemini 2.5 Flash, Gemini 2.5 Pro, VideoLLaMA 2, Qwen2.5-Omni-7B, video-SALMONN 2-7B, and Omni-Captioner-7B. We collect 500 pairs of captions generated by different models and use these pairwise comparisons as simulated matches in the Elo rating procedure. This human-based ranking serves as a reference to evaluate how closely Omni-Cloze scores align with actual human judgments. The hyperparameters applied in the Elo ranking system are summarized in Table 15.

Table 15: Hyperparameters of the Elo ranking system for arena-style evaluation.

| Hyperparameter | Value |
|---|---|
| Initial Elo Mean | 1,000 |
| Base of Logarithm | 10 |
| Scaling Factor | 400 |
| K-Factor | 32 |
| Simulated Matches | 500 |

#### D.1.2  MODALITY-WISE ANALYSIS

Figure 9 shows the relationship between Elo scores and Omni-Cloze accuracy across visual-only, (b) audio-only, and audio-visual modalities, which demonstrate strong positive correlations.

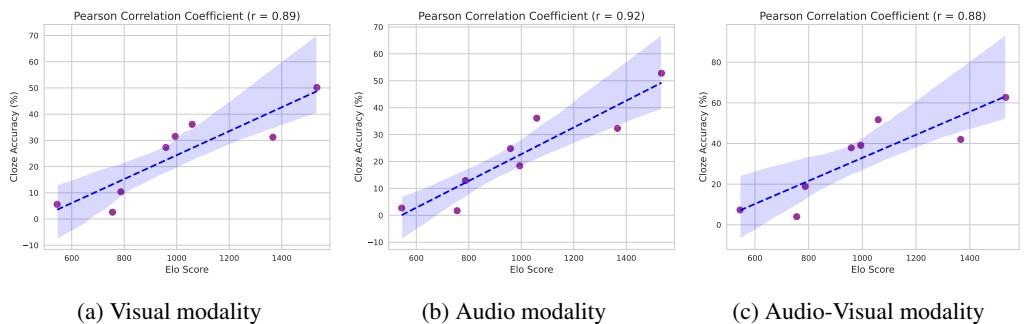

(a) Visual modality    (b) Audio modality    (c) Audio-Visual modality

Figure 9: Scatter plots showing the relationship between Elo scores and Omni-Cloze accuracy across different modalities: (a) visual, (b) audio, and (c) audio-visual.

## E  LIMITATIONS

Although this work explicitly explores hallucination issues in detailed captioning, a key limitation is that our evaluation cannot detect all hallucination types. In open-domain detailed captioning, we observe two main forms of hallucination: (1) The model correctly identifies the presence of a detail in the input but supplies incorrect information about it (content-level inaccuracy); (2) The model outputs content entirely unrelated to the given audio–visual input (irrelevant generation). Our proposed Omni-Cloze benchmark is able to capture and penalize the first type by adding a Not Given choice for each blank. However, the second type remains difficult to measure reliably because cases exist where the model predicts a detail that is in fact present in the audio–visual input but is missing from the ground truth reference. Developing robust methods to handle this issue is an important direction for future work in hallucination evaluation for omni detailed perception.

# F CASE STUDY

Table 16: Comprehensive error analysis for all cloze blanks in the PBA-League passage.

| # | Modality | Cue (short description) | Correct answer | Predicted answer | GT | Pred | ✓/✗ |
|---|---|---|---|---|---|---|---|
| 1 | Visual | Arena venue | Bayside Bowl | Bayside Bowl | C | C | ✓ |
| 2 | Visual | Team shown on graphic | L.A. X | L.A. X | B | B | ✓ |
| 3 | Visual | Shirt colour (Butturff) | red | red | B | B | ✓ |
| 4 | Visual | Ear with earring | left | left | D | D | ✓ |
| 5 | Visual | Event where he finished 2nd | U.S. OPEN | not given | C | E | ✗ |
| 6 | Auditory | First organised crowd chant | Jakob | L.A. X | D | C | ✗ |
| 7 | Auditory | Sound effect type | stinger | stinger | C | C | ✓ |
| 8 | Auditory | Secondary effect | whoosh | whoosh | D | D | ✓ |
| 9 | Audio-Visual | Bowler named on overlay | Jakob Butturff | Jakob Butturff | A | A | ✓ |
| 10 | Visual | Series score (semifinal lead) | 1–0 | not given | B | E | ✗ |
| 11 | Visual | Kingpins score (mid-game) | 117 | 117 | C | C | ✓ |
| 12 | Visual | Strike-Track ball speed | 18.6 MPH | not given | B | E | ✗ |
| 13 | Visual | Strike-Track rev-rate | 533 RPM | not given | B | E | ✗ |
| 14 | Auditory | Ball-rumble stereo position (1st shot) | dead centre | dead centre | A | A | ✓ |
| 15 | Auditory | Pinfall result | strike | strike | A | A | ✓ |
| 16 | Auditory | Handedness called | lefty | lefty | D | D | ✓ |
| 17 | Auditory | Player-of-year candidate named | Jason Belmonte | not given | D | E | ✗ |
| 18 | Visual | Fist-pump hand | right | not given | B | E | ✗ |
| 19 | Audio-Visual | Celebration gesture | high-fives | high-fives | B | B | ✓ |
| 20 | Visual | Audience accessories | Santa hats | not given | D | E | ✗ |
| 21 | Visual | "Coming Up Next" opponent | Portland | Portland | C | C | ✓ |
| 22 | Audio-Visual | Logo on Sherman's sleeve | STORM | Turbo | C | B | ✗ |
| 23 | Auditory | Crowd-chant word 1 | Sherman | Sherman | D | D | ✓ |
| 24 | Auditory | Crowd-chant word 2 | Sherman | Sherman | D | D | ✓ |
| 25 | Auditory | Crowd-chant word 3 | Sherman | Sherman | D | D | ✓ |
| 26 | Auditory | Stands song | Hey! Hey-hey! Goodbye! | Hey! Hey-hey! Goodbye! | C | C | ✓ |
| 27 | Audio-Visual | Leg kicked in follow-through | left leg | right leg | D | B | ✗ |
| 28 | Auditory | Ball-rumble channel location (2nd shot) | centre | not given | A | E | ✗ |
| 29 | Audio-Visual | Remaining pin state | wobbling | wobbling | C | C | ✓ |
| 30 | Auditory | Crowd reaction sound | groan | groan | B | B | ✓ |
| 31 | Auditory | Spare-leave call | ringing ten | not given | C | E | ✗ |
| 32 | Audio-Visual | Kingpins total after 8th frame | 146 | 136 | B | C | ✗ |

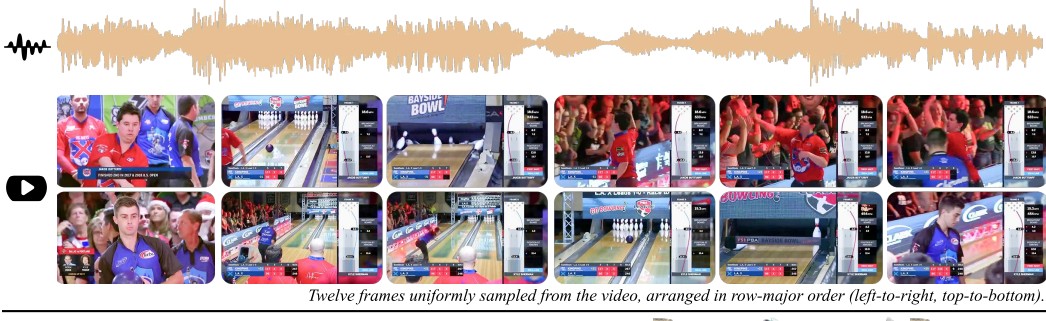

*Twelve frames uniformly sampled from the video, arranged in row-major order (left-to-right, top-to-bottom).*

**Omni-Captioner**                    📟 Visual    🎵 Auditory    🎬 Audio-Visual

The segment opens in a packed bowling arena at [📟 ✅ Bayside Bowl] (A: Thunderbowl Lanes | B: South Point Bowling Plaza | C: Bayside Bowl | D: AMF Riviera Lanes | E: not given), a PBA League semifinal between [📟 ✅ L.A. X] (A: Portland Lumberjacks | B: L.A. X | C: Motown Muscle | D: Dallas Strikers | E: not given) and the Kingpins. The first image is a medium close-up of Jakob Butturff in a bright [📟 ✅ red] (A: black | B: red | C: blue | D: green | E: not given) polo with a blue X over a bear shield, lips pressed and focused, a small silver earring in his [📟 ✅ left] (A: right | B: both | C: none | D: left | E: not given) ear. A blue-and-white lower-third flashes "JAKOB BUTTURFF" with "FINISHED 2ND IN 2017 & 2018 [📟 ✅ U.S. OPEN] (A: Masters | B: Tournament of Champions | C: U.S. OPEN | D: World Championship | E: not given)" beneath. A male commentator, in a clear broadcast tone, says: "opposite to 21, Jakob Butturff." From the crowd comes an organized chant of "[🎵 ❌ L.A. X] (A: Let's go | B: Butturff | C: L.A. X | D: Jakob | E: not given)" that swells beneath the call. A sharp electronic [🎵 ✅ stinger] (A: jingle | B: siren | C: stinger | D: drumroll | E: not given) with a [🎵 ✅ whoosh] (A: horn | B: clang | C: beep | D: whoosh | E: not given) punctuates the build as the camera tracks behind the first bowler. The stinger and whoosh signal imminent action just as [🎬 📟 ✅ Jakob Butturff] (A: Jakob Butturff | B: Norm Duke | C: Jason Belmonte | D: Kyle Sherman | E: not given) begins his approach on screen. The left overlay shows "Semifinals - L.A. X Lead [📟 ❌ not given] (A: 0-1 | B: 1-0 | C: 1-1 | D: 2-0 | E: not given)" with Kingpins [📟 ✅ 117] (A: 96 | B: 126 | C: 117 | D: 146 | E: not given) and L.A. X 96, while the Strike Track animates: his first-shot line overlays the ideal, showing [📟 ❌ not given] (A: 16.8 MPH | B: 18.6 MPH | C: 19.3 MPH | D: 20.1 MPH | E: not given) and [📟 ❌ not given] (A: 454 RPM | B: 533 RPM | C: 380 RPM | D: 600 RPM | E: not given). The ball's low rumble grows [🎵 ✅ dead center] (A: dead center | B: hard left | C: offstage | D: far right | E: not given) and then detonates into a clean, explosive [🎵 ✅ strike] (A: strike | B: spare | C: split | D: gutter | E: not given), sending all ten pins down and unleashing a roar from the crowd. Announcer 2 exclaims, "Ooh, what an angle that was to the pocket for the [🎵 ✅ lefty] (A: rookie | B: righty | C: veteran | D: lefty | E: not given)," then adds, "By the way, we've talked about [🎵 ❌ not given] (A: Pete Weber | B: Kyle Troup | C: Norm Duke | D: Jason Belmonte | E: not given) as player of the year candidate." Butturff pivots left, slams a downward [📟 ❌ not given] (A: left | B: right | C: both | D: no | E: not given)-handed fist pump, then strides to the rail for a gauntlet of [🎬 📟 ✅ high-fives] (A: chest bumps | B: high-fives | C: handshakes | D: hugs | E: not given) as red-tinted audience lighting flashes over [📟 ❌ not given] (A: beanies | B: cowboy hats | C: foam fingers | D: Santa hats | E: not given) and other holiday outfits. A "Coming Up Next: DALLAS vs [📟 ✅ PORTLAND] (A: DALLAS | B: SEATTLE | C: PORTLAND | D: HOUSTON | E: not given)" graphic with portraits appears. The rhythm resets for the Kingpins. A medium shot frames Kyle Sherman in a blue-and-black polo marked with a Turbo logo, [🎬 📟 ❌ Turbo] (A: Roto Grip | B: Turbo | C: STORM | D: Ebonite | E: not given) on the sleeve, a "500" patch, and a shielded Statue of Liberty. The crowd's buzz eases into a pocket of anticipation, and from the right side a chant of "[🎵 ✅ Sherman] (A: Belmo | B: Jakob | C: Portland | D: Sherman | E: not given), [🎵 ✅ Sherman] (A: Belmo | B: Jakob | C: Portland | D: Sherman | E: not given), [🎵 ✅ Sherman] (A: Belmo | B: Jakob | C: Portland | D: Sherman | E: not given)" rises. As he starts his approach, some in the stands sing "[🎵 ❌ not given] (A: We will rock you! | B: Defense! | C: Hey! Hey-hey! Goodbye! | D: Let's go! | E: not given)." The Strike Track updates: 19.3 MPH, 454 RPM, position at arrows 12.0, breakpoint board 4.2, while Sherman's [🎬 📟 ❌ right] (A: rear | B: right | C: front | D: left | E: not given) leg kicks high in the follow-through. The ball's rumble builds in the [🎵 ❌ not given] (A: center | B: rear | C: left channel | D: right channel | E: not given) and then cracks sharply; chaos scatters nine pins and leaves the back-right pin [🎬 📟 ✅ wobbling] (A: sliding | B: falling | C: wobbling | D: spinning | E: not given) but upright. A collective "Oh!" and then a [🎵 ✅ groan] (A: hiss | B: groan | C: laugh | D: whistle | E: not given) rolls through the venue. Commentator 2 reacts, "Wow, a violent [🎵 ❌ not given] (A: pocket seven | B: stone eight | C: ringing ten | D: solid nine | E: not given) against Kyle Sherman." The scoreboard updates the eighth frame with a 9 and a Kingpins total of [🎬 📟 ❌ 136] (A: 156 | B: 146 | C: 136 | D: 126 | E: not given) while L.A. X remains at 96. Sherman turns away, expression tight, disengaging without celebration, and the pinsetter machinery thrums as it resets the deck. Around the set, banners read PBA LEAGUE, BAYSIDE BOWL, and GO BOWLING, with sidebars looping CLARK logos, as the broadcast settles back into steady analysis and crowd murmur.

Figure 10: Visualization for all cloze blanks in the PBA-League passage.

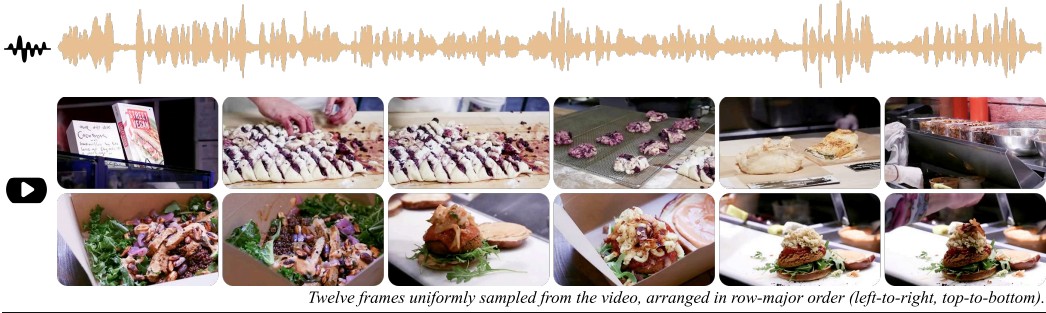

*Twelve frames uniformly sampled from the video, arranged in row-major order (left-to-right, top-to-bottom).*

---

**Omni-Captioner**                    📺 Visual      🎷 Auditory      📣📺 Audio-Visual

The video opens on a crisp static shot of a cookbook titled [📺✅ STREET VEGAN] (A: STREET VEGAN | B: GREEN KITCHEN | C: VEGAN STREET FOOD | D: PLANT POWER | E: not given), credited to [📺❌ not given] (A: Chloe Coscarelli | B: Isa Chandra Moskowitz | C: Adam Sobel | D: Bryant Terry | E: not given), beside a blocky white sign handwritten in [📺❌ not given] (A: black | B: green | C: red | D: blue | E: not given) ink that whimsically mentions "Soft Space-age [📺❌ not given] (A: Robot-chefs | B: Dog-men | C: Cat-people | D: Space Bears | E: not given)." Under this, a relaxed [🎷✅ jazz/lounge] (A: classical | B: rock | C: hip-hop | D: jazz/lounge | E: not given) music track plays as low ambient walla suggests a busy [🎷❌ not given] (A: library | B: office | C: gym | D: restaurant | E: not given) or cafe, with occasional [🎷❌ not given] (A: whistles | B: applause | C: footsteps | D: clinks | E: not given) of glasses. A warm, close-miked male voice in a [🎷❌ not given] (A: tenor-to-baritone | B: falsetto | C: monotone | D: soprano | E: not given) range begins, saying veganism can be "extra [🎷✅ delicious] (A: affordable | B: healthy | C: ethical | D: delicious | E: not given) and innovative," and the filler word "[🎷❌ not given] (A: you know | B: like | C: um | D: uh | E: not given)" pops up twice; a brief high-pitched [🎷❌ not given] (A: male | B: female | C: robotic | D: child | E: not given) laugh flares in the background. The scene cuts to a close-up of fair-skinned hands in a [📺❌ not given] (A: blue | B: white | C: gray | D: black | E: not given) long-sleeved garment working a braided dough that oozes a dark [📺❌ brown] (A: yellow | B: brown | C: purple | D: red | E: not given) chunky filling, and torn pieces are placed onto a [📺❌ not given] (A: ceramic | B: solid | C: mesh | D: wooden | E: not given) baking tray. A display counter appears with placards: one reads "SPINACH PIE WITH ROASTED [📺❌ not given] (A: GARLIC | B: MUSHROOMS | C: PEPPERS | D: ONION | E: not given) AND CAPERS," while another labels a round pastry "CURRIED [📺❌ not given] (A: POTATO | B: LENTIL | C: CHICKPEA | D: TOFU | E: not given) PUFF." As he lists peanut components, the visuals cut to a vibrant salad served in a brown [📣📺✅ take-out] (A: glass | B: ceramic | C: metal | D: take-out | E: not given) box while he says the peanuts are tossed with smoked chilies, a little dash of [🎷✅ agave nectar] (A: agave nectar | B: brown sugar | C: maple syrup | D: molasses | E: not given), and [🎷✅ tamari] (A: soy sauce | B: miso | C: mirin | D: tamari | E: not given). He rejects blandness with, "I'm not gonna use just basic, plain-ass [🎷✅ peanuts] (A: cashews | B: walnuts | C: peanuts | D: almonds | E: not given) that don't have any pizzazz; I wanna [🎷❌ season] (A: dress | B: hook | C: season | D: plate | E: not given) it up with pizzazz." The climactic build shows the proclamation "This is the Beast Mode Burger Deluxe" aligning with the burger being assembled on a [📣📺❌ not given] (A: steel | B: black | C: white | D: wooden | E: not given) cutting board: a bun bottom spread with glossy orange-tinted sauce, fresh [📺✅ arugula] (A: arugula | B: lettuce | C: spinach | D: kale | E: not given) layered on, then a thick, crisp brown patty, after which, immediately following the naming, a generous helping of a [📣📺❌ not given] (A: green | B: yellow | C: white | D: red | E: not given)-colored sauce is added. Sliced, [📺✅ translucent] (A: opaque | B: minced | C: translucent | D: charred | E: not given) pickled onions glisten as they are placed with tongs, and a scoop of creamy, pale [📺❌ not given] (A: potato salad | B: macaroni and cheese | C: coleslaw | D: risotto | E: not given) with flecks of herbs crowns the stack. A [📺✅ tattooed] (A: sleeved | B: tattooed | C: braceleted | D: bandaged | E: not given) forearm enters to sprinkle something crispy and dark over the top. As he says, "And it gets finished with some of this smoked chili coconut bacon," a gentle [📣📺❌ sprinkling] (A: sprinkling | B: sizzling | C: chopping | D: whirring | E: not given) sound rises while the crispy dark topping falls onto the mac and cheese. The clip cuts off abruptly mid-word during "[🎷❌ not given] (A: burger | B: bacon | C: beast | D: deluxe | E: not given)," with no fade. At the very start, while the cookbook and quirky sign hold the frame, the narration frames vegan food as "extra delicious and [📣📺✅innovative] (A: traditional | B: innovative | C: simple | D: spicy | E: not given)."

---

Figure 11: Visualization for all cloze blanks in the "Street Vegan" passage.

Table 17: Comprehensive error analysis for all cloze blanks in the "Street Vegan" passage.

| # | Modality | Cue (short description) | Correct answer | Predicted answer | GT | Pred | ✓/✗ |
|---|---|---|---|---|---|---|---|
| 1 | Visual | Cookbook title | STREET VE-GAN | STREET VE-GAN | A | A | ✓ |
| 2 | Visual | Author credited | Adam Sobel | not given | C | E | ✗ |
| 3 | Visual | Ink colour on sign | black | not given | A | E | ✗ |
| 4 | Visual | "Soft Space-age …" creature | Dog-men | not given | B | E | ✗ |
| 5 | Auditory | Background music style | jazz/lounge | jazz/lounge | D | D | ✓ |
| 6 | Auditory | Ambient location sound | restaurant | not given | D | E | ✗ |
| 7 | Auditory | Occasional background sound | clinks | not given | D | E | ✗ |
| 8 | Auditory | Narrator voice range | tenor-to-baritone | not given | A | E | ✗ |
| 9 | Auditory | Adjective used ("extra …") | delicious | delicious | D | D | ✓ |
| 10 | Auditory | Repeated filler word | like | not given | B | E | ✗ |
| 11 | Auditory | High-pitched laugh – voice type | female | not given | B | E | ✗ |
| 12 | Visual | Colour of long-sleeved garment | white | not given | B | E | ✗ |
| 13 | Visual | Colour of dark chunky filling | purple | brown | C | B | ✗ |
| 14 | Visual | Type of baking tray | mesh | not given | C | E | ✗ |
| 15 | Visual | Roasted ingredient on placard | GARLIC | not given | A | E | ✗ |
| 16 | Visual | Curried ingredient on placard | LENTIL | not given | B | E | ✗ |
| 17 | Audio-Visual | Salad served in … box | take-out | take-out | D | D | ✓ |
| 18 | Auditory | Sweetener named | agave nectar | agave nectar | A | A | ✓ |
| 19 | Auditory | Savoury seasoning named | tamari | tamari | D | D | ✓ |
| 20 | Auditory | Nut type criticised | peanuts | peanuts | C | C | ✓ |
| 21 | Auditory | Verb actually spoken ("… it up") | hook | season | B | C | ✗ |
| 22 | Audio-Visual | Cutting-board colour/material | white | not given | C | E | ✗ |
| 23 | Visual | Leafy green placed on burger | arugula | arugula | A | A | ✓ |
| 24 | Audio-Visual | Colour of sauce added after naming | red | not given | D | E | ✗ |
| 25 | Visual | Appearance of pickled onions | translucent | translucent | C | C | ✓ |
| 26 | Visual | Creamy topping item | macaroni and cheese | not given | B | E | ✗ |
| 27 | Visual | Forearm characteristic | tattooed | tattooed | B | B | ✓ |
| 28 | Audio-Visual | Sound during topping | sprinkling | sprinkling | A | A | ✓ |
| 29 | Auditory | Word cut off at the end | bacon | not given | B | E | ✗ |
| 30 | Audio-Visual | Descriptor paired with "extra delicious" | innovative | innovative | B | B | ✓ |

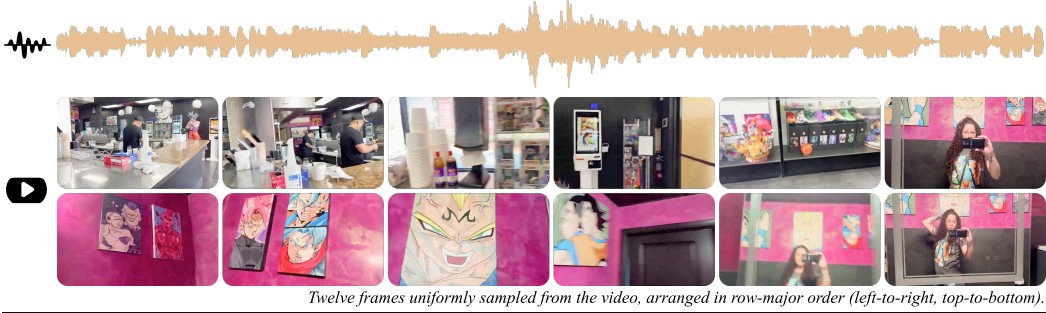

*Twelve frames uniformly sampled from the video, arranged in row-major order (left-to-right, top-to-bottom).*

**Omni-Captioner**              🗄 Visual  🎧 Auditory  📣🗄 Audio-Visual

A vertical smartphone video tours a restaurant entirely themed around [🗄✅ Dragon Ball Z] (A: One Piece | B: Naruto | C: Dragon Ball Z | D: Pokémon | E: not given), while an energetic electronic soundtrack pulses beneath. The music opens with a classic [🎧✅ four-on-the-floor] (A: swing | B: waltz | C: four-on-the-floor | D: half-time | E: not given) kick pattern and bright synth [🎧❌ not given] (A: glissandos | B: arpeggiations | C: tremolos | D: strums | E: not given) driving a glossy melody. As the camera glides past a stainless prep area to a granite-patterned counter, two employees in black T-shirts and caps work methodically—one cap worn forward, the other [🗄✅ backward] (A: forward | B: inside out | C: backward | D: sideways | E: not given)—with a [🗄❌ not given] (A: white | B: red | C: blue | D: green | E: not given) spray bottle nearby, sneeze guards separating stations, and cloud decor hanging from a [🗄❌ not given] (A: red | B: white | C: blue | D: black | E: not given) ceiling in front of a large monochrome [🗄❌ not given] (A: Piccolo | B: Vegeta | C: Frieza | D: Goku | E: not given) face mural. A hard cut reveals tall glass cases by a window looking out to a parking lot, and later a whip-pan catches a male customer turning away; he says [📣🗄❌ not given] (A: "hey" | B: nothing | C: a quiet laugh | D: "excuse me" | E: not given) as he leaves the frame. The display sweep shows collectible figures and boxes, including a figurine of [🗄❌ not given] (A: Gohan | B: Vegeta | C: Krillin | D: Cell | E: not given) with crossed arms and boxed [🗄❌ not given] (A: Hot Wheels | B: Figma | C: Nendoroid | D: Funko Pop | E: not given) figures below, plus a game box that reads [🗄✅ CLUE] (A: MONOPOLY | B: LIFE | C: RISK | D: CLUE | E: not given) in bold white at the top and a replica [🗄❌ not given] (A: Four | B: Seven | C: Five | D: Three | E: not given)-Star Dragon Ball. A pan lands on a life-sized Ultra Instinct statue with [🗄❌ not given] (A: black | B: gold | C: blue | D: silver | E: not given) hair beside two sleek kiosks topped with [🗄❌ not given] (A: blue | B: red | C: green | D: purple | E: not given) indicator lights. As the soundtrack builds, a rising whoosh [🎧✅ riser] (A: gong hit | B: stinger | C: riser | D: vinyl scratch | E: not given) swells, and the compression creates a rhythmic [🎧✅ pumping] (A: pumping | B: crackling | C: fluttering | D: ringing | E: not given) against the beat. A hard cut drops the viewer into a vivid restroom with magenta-on-black walls and an LED mirror; the person filming appears via her reflection. She speaks while visible in the [📣🗄✅ mirror] (A: mirror | B: doorway | C: poster | D: window | E: not given), her General [🎧❌ not given] (A: Australian | B: British RP | C: American | D: Southern drawl | E: not given) accent clear as she says, "Alright, so I'm in the bathroom. Pretty [🎧❌ cool] (A: wild | B: strange | C: funny | D: cool | E: not given)." There is a brief [🎧✅ pause] (A: sigh | B: laughter | C: cough | D: pause | E: not given) between "bathroom" and "pretty wild," and a faint [🎧❌ not given] (A: ringtone | B: clapping | C: footsteps | D: rustling | E: not given) at the start of her speech. No [🎧❌ not given] (A: faucet drip | B: hand dryer | C: sink | D: toilet | E: not given) sounds are heard. During this mirror shot, the music's studio sheen dips and sounds distant and reverberant, matching the small hard-surfaced room. Right after she finishes, the pristine track slams back in via a precise [🎧✅ hard edit] (A: gradual fade-in | B: hard edit | C: slow crossfade | D: long echo | E: not given), launching a build that leads to the drop with a powerful bassline and a classic [🎧✅ supersaw] (A: Rhodes | B: electric guitar | C: supersaw | D: acoustic piano | E: not given) lead timbre. Short, high-pitched processed accents—"[🎧❌ not given] (A: Yo! | B: Heya! | C: Go! | D: Hey! | E: not given)!"—cut through the mix, and they occur while the camera remains in the [📣🗄✅ bathroom] (A: bathroom | B: kitchen | C: storage room | D: dining area | E: not given). Near the end of the bathroom segment, she [📣🗄✅ adjusts her hair] (A: sets the phone down | B: opens the faucet | C: adjusts her hair | D: claps her hands | E: not given), and a brief fabric sound coincides with the motion. Throughout, the track drives in [🎧✅ 4/4] (A: 3/4 | B: 7/8 | C: 4/4 | D: 5/4 | E: not given) time and finally ends with an [🎧✅ abrupt stop] (A: gradual fade-out | B: tape slowdown | C: abrupt stop | D: extended reverb tail | E: not given), not a fade. Meanwhile, the woman's look in the mirror includes glasses, visible tattoos, a silver cross necklace, and an anime T-shirt, and while delivering the line she holds her smartphone [📣🗄❌ not given] (A: diagonally | B: upside down | C: horizontally | D: vertically | E: not given).

Figure 12: Visualization for all cloze blanks in the Dragon-Ball-Z-themed-restaurant passage.

Table 18: Comprehensive error analysis for all cloze blanks in the Dragon-Ball-Z-themed-restaurant passage.

| # | Modality | Cue (short description) | Correct answer | Predicted answer | GT | Pred | ✓/✗ |
|---|----------|-------------------------|----------------|------------------|----|----|-----|
| 1 | Visual | Restaurant theme | Dragon Ball Z | Dragon Ball Z | C | C | ✓ |
| 2 | Auditory | Kick-drum pattern | four-on-the-floor | four-on-the-floor | C | C | ✓ |
| 3 | Auditory | Synth ornament | arpeggiations | not given | B | E | ✗ |
| 4 | Visual | Second employee's cap orientation | backward | backward | C | C | ✓ |
| 5 | Visual | Colour of spray bottle | white | not given | A | E | ✗ |
| 6 | Visual | Ceiling colour with cloud décor | black | not given | D | E | ✗ |
| 7 | Visual | Character on monochrome face mural | Goku | not given | D | E | ✗ |
| 8 | Audio-Visual | Customer utterance while exiting | (nothing) | not given | B | E | ✗ |
| 9 | Visual | Arms-crossed figurine | Vegeta | not given | B | E | ✗ |
| 10 | Visual | Type of boxed figures | Funko Pop | not given | D | E | ✗ |
| 11 | Visual | Board-game box shown | CLUE | CLUE | D | D | ✓ |
| 12 | Visual | Replica Dragon Ball (number of stars) | Four-Star | not given | A | E | ✗ |
| 13 | Visual | Ultra-Instinct statue hair colour | silver | not given | D | E | ✗ |
| 14 | Visual | Kiosk indicator-light colour | blue | not given | A | E | ✗ |
| 15 | Auditory | Transition SFX type | riser | riser | C | C | ✓ |
| 16 | Auditory | Side-chain / level effect | pumping | pumping | A | A | ✓ |
| 17 | Audio-Visual | Surface that shows filmer (bathroom) | mirror | mirror | A | A | ✓ |
| 18 | Auditory | Speaker's accent | American | not given | C | E | ✗ |
| 19 | Auditory | Adjective in "Pretty …" | wild | not given | A | E | ✗ |
| 20 | Auditory | Intentional pause after "bathroom" | pause | pause | D | D | ✓ |
| 21 | Auditory | Faint sound at speech start | rustling | not given | D | E | ✗ |
| 22 | Auditory | Sound explicitly absent ("No … heard") | toilet | not given | D | E | ✗ |
| 23 | Auditory | Music returns via | hard edit | hard edit | B | B | ✓ |
| 24 | Auditory | Lead-synth timbre | supersaw | supersaw | C | C | ✓ |
| 25 | Auditory | Short processed vocal accent | "Hey!" | not given | D | E | ✗ |
| 26 | Audio-Visual | Room shown during vocal accents | bathroom | bathroom | A | A | ✓ |
| 27 | Audio-Visual | Action: filmer adjusts … | her hair | her hair | C | C | ✓ |
| 28 | Auditory | Time signature of track | 4/4 | 4/4 | C | C | ✓ |
| 29 | Auditory | Track ending style | abrupt stop | abrupt stop | C | C | ✓ |
| 30 | Audio-Visual | Orientation of phone in her hand | horizontally | not given | C | E | ✗ |

