# OpenReview forum: "Omni-Captioner: Data Pipeline, Models, and Benchmark for Omni Detailed Perception"
_ICLR.cc/2026/Conference — ICLR 2026 Poster_

### Official Review · Reviewer_AErd · 2025-10-23

**Soundness:** 3
**Presentation:** 4
**Contribution:** 3
**Rating:** 8
**Confidence:** 3

**Summary:**

The paper introduces Omni-Detective, an agentic data generation pipeline that uses iterative Query-Observation cycles to produce high-quality, low-hallucination multimodal captions. Based on this data, the authors train Audio-Captioner and Omni-Captioner models for audio and audio-visual detailed perception, respectively, and propose Omni-Cloze, a novel cloze-style benchmark for evaluation. Experimental results show that Omni-Captioner outperforms existing open-source models on benchmarks like VDC and video-SALMONN 2, achieving a strong balance between detail and hallucination.

**Strengths:**

# Strengths:
(+) Innovative data generation：Omni-Detective’s iterative approach with tool-calling effectively reduces hallucination, providing a novel solution for generating high-quality multimodal data.

(+) Superior model performance：Omni-Captioner sets new state-of-the-art results on VDC and achieves the best detail-hallucination trade-off on video-SALMONN 2, surpassing most open-source models.

(+) Robust evaluation framework：Omni-Cloze offers a comprehensive, efficient, and reliable cloze-style evaluation across audio, visual, and audio-visual modalities.

**Weaknesses:**

# Weakness:
(-) The reliance on a single detective module, while effective, lacks the depth of insight offered by multi-agent systems, potentially limiting its innovation in the generative AI field.

(-) The paper lacks experimental data on the computational cost per iteration, leaving the method’s feasibility in resource-constrained environments unclear.

(-) The paper does not specify how to assess the detective agent’s performance, raising concerns that poor agent output could result in longer, meaningless text rather than valuable information.

**Questions:**

# Questions:
1. Can Omni-Detective maintain low hallucination rates during multi-round iterations in highly complex scenarios?
2. Are there plans to evaluate Omni-Captioner’s performance on devices with limited computational resources?
3. How can specific metrics be designed to quantify the quality of information extracted by the detective agent in each round?
4. Is there a mechanism to correct errors if the detective agent misidentifies information and mitigate subsequent impacts?
5. Can the Omni-Cloze benchmark be extended to evaluate longer videos or more complex multimodal interactions?

---

> ### Author Response · Authors · 2025-11-23
> **General Response**
>
> Dear Reviewer AErd,
>
> Thank you very much for your review and high rating. We are delighted that you recognized the innovation of Omni-Detective, the superior performance of Omni-Captioner, and the robustness of the Omni-Cloze evaluation framework.
>
> We will now address your concerns and questions in detail.

---

> ### Author Response · Authors · 2025-11-23
> **Response 1**
>
> ## W1: Question on the detective module.
> We appreciate this perspective and must clarify that the Omni-Detective is not a single detection module but an agentic orchestration system designed to manage multiple diverse perception tools. The innovation lies precisely in the agent's ability to orchestrate and verify across these tools iteratively.
>
> The core insights and novelty derived from this design are:
> - **Adaptive Scheduling (Minimizing Human Prior)**: The agent autonomously decides which tool to call next based on current evidence, rather than relying on a fixed, pre-defined sequence, thereby maximizing detail extraction with minimal human prior knowledge.
> - **Divide and Conquer (Simplifying Complex Tasks)**: The complex task of generating one single, hyper-detailed caption is broken down into multiple, simpler tasks achievable by specialized tool models. This utilizes a tool-calling mechanism to decompose, which is an effective strategy for complex data construction.
> - **Self-Correction and Self-Consistency (Enhancing Detail, Reducing Hallucination)**: The multi-turn Query-Observation cycle acts as a verification loop. If new information from an Observer conflicts with previous claims, the Detective Agent can adaptively launch further investigative turns to cross-check the evidence, thus explicitly targeting the decoupling of detail gain from hallucination growth.
>
> ## W2: Question on computational costs.
> We acknowledge this difficulty, as an agentic data generation pipeline, precisely estimating the total computational cost (both time and resources) is inherently complex for several reasons:
> 1. **Adaptive Step Budget**: The agent adaptively schedules its investigation steps, and the required number of turns is variable depending on the content complexity of the video clip (the agent may early stop or use its full budget of steps).
> 2. **Tools/APIs Variability**: Specialist tools are called via API or local deployment. The latency and throughput are dependent on external service loads and concurrency settings, making stable time estimation difficult.
>
> For our setting (as noted in Appendix A.1.1), the average generation time per high-quality data sample is approximately 12 minutes (with a concurrency level of 100), resulting in around 7 seconds per sample. We view this upfront cost as necessary and highly efficient for producing high-fidelity, low-hallucination ground truth data.
>
> ## W3 & Q3: Question on the detective agent’s performance.
> We performed a comprehensive analysis in the paper to ensure Omni-Detective generates valuable information:
> 1. **Quantitative Metrics (Figure 6)**: We tracked Detail Rate, Not-Given Rate, and Hallucination Rate across the Agent's steps. This serves as a quantitative metric to show that as Omni-Detective iterates, it reliably increases useful information while stabilizing errors.
> 2. **Downstream Utility (Table 5)**: The performance gains in the cascade QA evaluation demonstrate that the captions generated by Omni-Detective are richer and more useful for complex downstream reasoning tasks.
> 3. **Qualitative Verification (New Analysis)**: Furthermore, we performed a manual error analysis (as discussed with Reviewer 1S9o) showing that out of 388 sentences analyzed, 346 were perfectly correct, demonstrating that Omni-Detective successfully avoids generating long, meaningless text.

---

> ### Author Response · Authors · 2025-11-23
> **Response 2**
>
> ## Q1: Question on multi-round iterations.
> Yes, our analysis in Figure 6 shows that Omni-Detective maintains a stable and low hallucination rate. The hallucination rate drops sharply initially and then converges relatively early (around step 5-6), remaining low despite subsequent iterations that continue to extract details (Detail Rate keeps climbing). This convergence implies Omni-Detective reaches an effective ceiling in eliminating incorrect claims (which are the capacity constraints of the tools).
>
> ## Q2: Question on limited computational resources.
> We acknowledge that models for embodied AI and edge devices are a promising direction. Our current work focuses on fundamental research advancements in the data pipeline, training methods, and evaluation protocols for detailed perception. We will monitor and contribute to the rapidly growing field of device-side model optimization in our future research.
>
> ## Q4: Question on the mechanism to correct errors in Omni-Detective.
> Yes, the error correction and conflict resolution loop is a key mechanism of Omni-Detective. For instance, as partially visualized in Figure 3:
> 1. In an early turn, the detective agent might query the visual stream and hypothesize about the presence of "multiple people cooking".
> 2. In a subsequent turn, the detective agent calls ASR and finds the speech content suggests all narration is "from chef Adam Sobel".
> 3. The detective agent then integrates this cross-modal evidence and concludes the final description refers to a "solo presentation by chef Adam Sobel", effectively correcting the initial, potentially hallucinated, visual inference.
>
> ## Q5:  Question on extending evaluation scope.
> Here we clarify the task definition. Detailed captioning is distinct from Dense video captioning. Detailed captioning focuses on maximizing the information density and fidelity within a single clip. Because the sheer volume of fine-grained details would increase exponentially with video length, long videos are inherently unsuitable for this specific task of maximizing detailed description. Our focus is on a comprehensive, fine-grained understanding of short video segments, which we clarify at Line 112.
>
> ------------
>
> ## Conclusion
>
> We believe these thorough explanations and justifications comprehensively address your concerns and validate the soundness of our work. We sincerely hope you will consider increasing your rating and confidence score. Thank you for your time and expertise.

---

### Official Review · Reviewer_1S9o · 2025-10-29

**Soundness:** 2
**Presentation:** 2
**Contribution:** 2
**Rating:** 4
**Confidence:** 3

**Summary:**

This work proposes Omni-Detective, an agent-based data generation pipeline that automatically produces highly detailed multimodal data with minimal hallucination by integrating tool-calling. Specifically, Omni-Detective interacts with diverse tools, including MLLMs, OCR, and ASR, to extract information from videos and summarize detailed perceptions. Using this information, the LLM integrates all perceptions to generate detailed captions, which can later be used for question–answer tasks. Two captioning models, Audio-Captioner and Omni-Captioner, are trained as part of Omni-Detective.
To further validate detailed perception capabilities, the authors introduce Omni-Cloze, a cloze-style evaluation dataset for detailed audio, visual, and audio-visual captioning. Experimental results show that Omni-Detective outperforms existing methods on both existing benchmarks and the Omni-Cloze dataset.

**Strengths:**

- Designing Omni-Detective is a valid and practical approach to fully utilize the perceptual capabilities of existing models for detailed captioning.
- Introducing the Omni-Cloze evaluation dataset for detailed captioning is valuable to the community. It enables cloze-style evaluation, addressing current benchmark limitations that require multiple LLM calls for assessment.

**Weaknesses:**

- Several important details are missing. For instance:
  - Which dataset is used for Figure 2 and how are hallucinations detected in long captions?
  - Which model serves as the detective agent in Figure 3?
  - What metrics are used in Table 3?
  - Which datasets are used for joint training and audio-only training?
  - How is the detail rate evaluated in L406?

- Although AudioCaps or Clotho are not specifically designed for detailed captioning, they can still be used to evaluate audio captioning capabilities. However, they are not included in the experiments.

- In Figure 4, the Omni-Cloze samples appear to include audio-only, visual-only, and audio-visual QAs. In Table 4, for the audio-only model, was the evaluation conducted only on audio-only QAs within Omni-Cloze?

- Error analysis of the data generation pipeline is missing. Even with human validation, such analysis is essential to assess the robustness and reliability of the generated dataset.

- For training Omni-Captioner (and Audio-Captioner), is full finetuning reallly effective? Why not LoRA?

**Questions:**

### Questions and Suggested Experiments

- Although AudioCaps and Clotho are not targeted for detailed captioning, evaluating on these datasets would help assess the model’s audio captioning capability.

- Since the data generation pipeline involves automation, conducting error analysis after human verification would clarify its robustness.

- Ablation on each modality to examine whether the audio module is necessary for the omni benchmark will help to understand that the model fully exploit all the modalities, and the benchmark requires all the modalities.

### Minor Questions and Suggestions

- In L606, VGGSound was used for training the audio-captioner. Does VGGSound provide ground-truth captions for training?

---

> ### Author Response · Authors · 2025-11-23
> **General Response**
>
> Dear Reviewer 1S9o,
>
> Thank you for your constructive review and for recognizing the value of the Omni-Detective pipeline and the utility of the Omni-Cloze evaluation benchmark. Your questions are insightful and target important details that we are happy to clarify.
>
> Below is our point-by-point response addressing your weaknesses and questions.

---

> ### Author Response · Authors · 2025-11-23
> **Response 1**
>
> ## W1: Clarification of details.
> 1. **About Figure 2**: We use detailed captions from Gemini 2.5 Pro and test on Omni-Cloze in Figure 2. We mention the hallucination of "the proportion of mentioned details that are factually incorrect despite being noticed by the model" at Line 66. In all our evaluations (including Omni-Cloze), Detail Rate is defined as the successful prediction of details (Accuracy), Hallucination Rate is defined as giving an incorrect specific answer instead of choosing "Not Given," and Not-Given Rate is the proportion of details the model fails to mention. These three sum to 100%.
> 2. **About Detective Agent Model**: We apologize for the omission. The Detective Agent used in the Omni-Detective pipeline is gpt-4.1-2025-04-14, which we will specify in Appendix A.1.1.
> 3. **About Table 3 Metrics**: Table 3 presents cascade evaluation where the model first generates detailed captions, which are subsequently used to perform downstream question-answering tasks, which we clarify at Line 228 and Line 290. The metrics are standard metrics for each respective Audio-QA or Audio-Visual-QA benchmark, with most of them are Accuracy (%).
> 4. **About Datasets for Training**: As described in Appendix A.1.1, the models are trained on 55k audio-only samples (sourced from VGGSound) for Audio-Captioner, and 15k audio-visual samples (sourced from Fine Video) for Omni-Captioner.
> 5. **About Detail Rate in L406**: In all our evaluations (including Omni-Cloze), Detail Rate is defined as the successful prediction of details (Accuracy), Hallucination Rate is defined as giving an incorrect specific answer instead of choosing "Not Given," and Not-Given Rate is the proportion of details the model fails to mention. These three sum to 100%. We will ensure unified terminology in the revision.
>
> ## W2 & Q1: Why excluding AudioCaps and Clotho.
> We chose to omit audio captioning benchmarks like AudioCaps and Clotho **because their ground-truth annotations are too short and rely on older metrics** (e.g., BLEU, METEOR, CIDEr), which we clarify at Line 134. Our goal is to evaluate fine-grained detailed perception.  For context, a detailed caption from our model for an AudioCaps clip is substantially richer:
>
> - **Audio**: The first example from AudioCaps(https://audiocaps.github.io/supp/).
> - **Ground Truth**: a man talking as another person whistles while water trickles on a hard surface in the background
> - **Audio-Captioner Prediction**: The audio clip is a 10-second, professionally produced sound piece designed to evoke tension and psychological unease. It opens with a steady, high-pitched, synthetic whistling tone that is unwavering and monophonic, immediately establishing a sense of alertness. Layered over this is a male voice, speaking in a calm, measured, and slightly detached tone in Standard American English. His words—"As the source gets closer. You begin to panic inside, but remain stone on the outside. As soon as it began, the whistling suddenly ceased"—directly describe the listener’s experience, creating an immersive, first-person narrative. The background features subtle, low-frequency hum and a faint, rhythmic ticking reminiscent of a ticking clock, further heightening the atmosphere of suspense and impending threat. The sound environment is dry and close, with no natural reverberation or ambient noise, suggesting a controlled studio or digitally constructed setting. The audio is clear and well-balanced, with the narration slightly forward in the mix and the whistling serving as a constant, tense undercurrent. There are no musical elements, no additional voices, and no environmental context clues, reinforcing the sense of psychological isolation. The style and structure strongly suggest that this is an excerpt from a larger work—likely a psychological thriller, horror, or experimental audio drama—designed to immerse the listener in a moment of suspense and internal conflict. The deliberate pacing, controlled delivery, and precise sound design all contribute to a mood of anxiety, tension, and unresolved threat.
>
> Evaluating our model on short captions using low-sensitivity metrics would **fail to reflect its true detailed perception capability**, and using Omni-Cloze is one of the core contribution of this work.
>
> ## W3: Question of Omni-Cloze for the audio-only model.
> The Omni-Cloze evaluation is designed to be modality-complete. For the audio-only model evaluation (Table 4a, Audio-Captioner), we conducted the assessment only on the audio-only subset of cloze questions (for both numerator and denominator calculation).

---

> ### Author Response · Authors · 2025-11-23
> **Response 2**
>
> ## W4 & Q2: Manual error analysis of the data generation pipeline.
> We appreciate this valuable suggestion. Ensuring the robustness of the automated pipeline is critical. We addressed this in the paper through the analysis of Omni-Detective in Section 6.2, specifically:
> 1. **Figure 6** analyzes the trends of detail rate, not-given rate, and hallucination rate as the number of Omni-Detective steps increases, showing convergence behaviors.
> 2. **Table 5** provides a cascade evaluation demonstrating the downstream QA performance improvements directly by the Omni-Detective pipeline.
>
> **To further address your concern regarding qualitative robustness, we conducted a rigorous human-verified error analysis** during the rebuttal. We randomly sampled 20 detailed captions generated by Omni-Detective (comprising a total of 388 sentences) and manually verified them against the raw audio-visual content.
> We classified segments into four categories: Factual Errors/Hallucinations (incorrect information), Vague/Insignificant Errors (technically present but imprecise or insignificant), Stylistic Improvements (correct but could be phrased better), and Correct. We further traced the source of errors to either the Video or Audio modality.
>
> | **Error Type**                   | **Visual** | **Audio** | **Total** |
> |----------------------------------|------------|-----------|-----------|
> | **Factual Errors/Hallucination** | 14         | 1         | 15        |
> | **Vague/Insignificant Errors**   | 13         | 2         | 15        |
> | **Stylistic Improvements**       | 8          | 4         | 12        |
> | **Correct Sentences**            | -          | -         | 346       |
>
> ### Detailed Observations:
> 1. **Visual Dominance in Hallucinations**: Almost all of the factual errors stemmed from the visual modality, while the audio modality remained robust.
> 2. **Factual Errors Breakdown**: These were primarily spatial/directional errors (8) and confabulation (3), with the remainder being miscellaneous identification errors.
> 3. **Minor Errors Breakdown**: These included imprecise color descriptions (4), missing OCR punctuation or insignificant visual details (4). On the audio side, errors were due to inherent ambiguity in the source file, such as unclear ASR (1) or indistinct background music (1).
> 4. **High Reliability**: Out of 388 sentences, 346 were perfectly correct, and an additional 12 only required stylistic smoothing (e.g., grammar tweaks or sentence flow adjustments) without containing factual errors.
>
> This analysis confirms that Omni-Detective is highly effective at producing detailed captions. And the human-verified results align with Hallucination Rate in Omni-Cloze (Around 3.5% in Figure 6). As rigorous manual verification of detailed long-form captions is labor-intensive (over 30 minutes per sample), we limited this study to 20 samples for the rebuttal. We commit to expanding this human-verified error analysis on a larger scale in the final version of the paper.
>
> ## W4: Why full fine-tuning.
> In scenarios where the amount and quality of data are considered sufficient (as with our Omni-Detective dataset) and computational resources allow (we used 8 x A100 80GB GPUs), full fine-tuning is typically preferred over LoRA as it offers the highest potential performance gains by optimizing all parameters, leading to better performance.

---

> ### Author Response · Authors · 2025-11-23
> **Response 3**
>
> ## Q3: Ablation on each modality.
> You ask whether the model fully exploits all modalities and whether the benchmark requires them. Our results strongly confirm this necessity.
> We ran an ablation by generating detailed captions using only single-modality inputs (Visual-Only, Audio-Only) and evaluated them against the respective modality splits in Omni-Cloze.
>
> | **Input Modality**      | **Visual Part Acc% ($\uparrow$)** | **Audio Part Acc% ($\uparrow$)** | **AV Part Acc% ($\uparrow$)** | **Total Acc% ($\uparrow$)** |
> |-------------------------|-----------------------------------|----------------------------------|-------------------------------|-----------------------------|
> | **AV (Omni-Captioner)** | 50.2                              | 52.8                             | 62.7                          | 53.5                        |
> | **Visual-Only**         | 43.2                              | 10.9                             | 30.6                          | 28.5                        |
> | **Audio-Only**          | 3.1                               | 42.4                             | 20.8                          | 21.4                        |
>
> The results show that single-modality input fails on another modality and performs poorly on AV modality, demonstrating that the model fully exploits all modalities and that the Omni-Cloze benchmark requires the integration of all modalities for peak performance.
>
> ## Q4: Question on VGGSound Caption Usage
>
> We did not use the original ground-truth captions from VGGSound or Fine Video for training. We used these datasets only as the source of raw audio/video clips, upon which we applied the Omni-Detective pipeline to generate new, high-fidelity detailed captions. This decision was based on the fact that the original captions were insufficient for detailed perception tasks.
>
> ------------
>
> ## Conclusion
>
> We believe these comprehensive explanations and the addition of the critical ablation experiments further strengthen the integrity and claims of our work. We sincerely hope these revisions and clarifications fully address your concerns and motivate you to increase your rating.

---

> ### Author Response · Authors · 2025-11-27
> **Promote Discussion**
>
> Dear Reviewer 1S9o,
>
> We sincerely appreciate the time and expertise you invested in reviewing our paper. We have carefully addressed all concerns raised in the reviews and have provided detailed point-by-point responses. As the discussion period is now underway, we would be grateful if you could review our responses and post your comments. We are committed to engaging to enhance the quality of our work.
>
> Thanks again for your continued attention.
>
> Best regards,
>
> The Authors

---

### Official Review · Reviewer_WAFR · 2025-10-31

**Soundness:** 4
**Presentation:** 4
**Contribution:** 4
**Rating:** 8
**Confidence:** 4

**Summary:**

This paper introduces Omni-Detective, an agentic data generation pipeline designed to enhance fine-grained multimodal perception in Omni Language Models (OLMs). It produces highly detailed yet low-hallucination audio–visual data and supports training of Audio-Captioner and Omni-Captioner, which achieve performance comparable to closed-source models (ex, Gemini). The authors also propose Omni-Cloze, a new cloze-style benchmark for evaluating detailed multimodal captioning across audio, visual, and audio–visual domains.

**Strengths:**

- The paper is well written and well structured, clearly presenting the motivation, methodology, and results.

- It achieves competitive or superior performance to closed-source models (e.g., Gemini 2.5 Flash and Pro) across multiple benchmarks.

- The experimental setup, dataset description, and evaluation protocol are presented in exceptional detail, ensuring transparency and reproducibility.

**Weaknesses:**

There are no major weaknesses identified in the paper. The overall work is solid and well executed; only a few minor points of curiosity or clarification remain, which are addressed in the Questions section below.

**Questions:**

- In Table 2, why are proprietary models such as Gemini 2.0 and 2.5 not included in the comparison, despite being used as baselines in other sections? Were these results unavailable or intentionally omitted for consistency?

- Since the core models are fine-tuned from existing architectures, what specific aspects of Omni-Detective or Omni-Captioner represent the novel contribution beyond data quality improvements?

- Are there plans to release the generated dataset and benchmark publicly, and if so, will accompanying annotation or tool-calling logs be shared for transparency?

- When conducting these experiments, how sensitive are the results to hyperparameters such as gradient accumulation, number of training epochs, or learning rate? Even with a strong data pipeline like Omni-Detective, it would be valuable to understand how much small tuning variations influence the final outcomes, based on the authors’ experience.

---

> ### Author Response · Authors · 2025-11-23
> **General Response**
>
> Dear Reviewer WAFR,
>
> Thank you for your very positive and constructive review. We are delighted that you found the overall work to be solid, well-executed, and without major weaknesses, and that you recognized the exceptional detail provided for transparency and reproducibility.
>
> We will now address your specific questions for clarification.

---

> ### Author Response · Authors · 2025-11-23
> **Response**
>
> ## Q1: Comparison with proprietary models.
> The primary reason for the initial exclusion of Gemini 2.0 and 2.5 results in Table 2 was due to the high computational cost and the non-standardized evaluation protocol of the VDC benchmark. To address your concern directly and enhance the completeness of Table 2, we have now run the evaluation for the Gemini series on VDC.
>
> | **Model**   | **Acc% ($\uparrow$)** | **Score ($\uparrow$)** |
> |--------------------------|---------------------------|----------------------------|
> | **Omni-Captioner** | **55.02**                | **2.75**                   |
> | **Gemini 2.5 Flash**    | 51.05                    | 2.51                       |
> | **Gemini 2.5 Pro**    | 54.21                    | 2.68                       |
>
> The results confirm that Omni-Captioner sets a new state-of-the-art on VDC, surpassing both Gemini 2.5 Flash and Gemini 2.5 Pro. We will update Table 2 to include these essential proprietary baselines in the final version.
>
> ## Q2: Contribution beyond data quality improvements.
> While data quality is a major outcome, the novelty of our contribution is systematic, addressing the "co-growth" challenge across the entire research stack including data generation, model training, and evaluation. For Omni-Detective, This is a conceptual shift from static, human-prompted generation to adaptive, multi-step, tool-calling investigation. The novelty lies in the process:
> - **Adaptive Scheduling (Minimizing Human Prior)**: The agent autonomously decides which tool to call next based on current evidence, rather than relying on a fixed, pre-defined sequence, thereby maximizing detail extraction with minimal human prior knowledge.
> - **Divide and Conquer (Simplifying Complex Tasks)**: The complex task of generating one single, hyper-detailed caption is broken down into multiple, simpler tasks achievable by specialized tool models. This utilizes a tool-calling mechanism to decompose, which is an effective strategy for complex data construction.
> - **Self-Correction and Self-Consistency (Enhancing Detail, Reducing Hallucination)**: The multi-turn Query-Observation cycle acts as a verification loop. If new information from an Observer conflicts with previous claims, the Detective Agent can adaptively launch further investigative turns to cross-check the evidence , thus explicitly targeting the decoupling of detail gain from hallucination growth.
>
> ## Q3: Questions about open-source.
> We are fully committed to open-sourcing our work to facilitate further research in omni detailed perception. The code for the Omni-Detective data generation pipeline and the trained models have been released (we are unable to provide them during the double-blind review phase). The Omni-Cloze benchmark is currently undergoing final quality checks and bias alignment. We plan to release the benchmark, the Omni-Detective tool-calling trajectories alongside the final version of the paper.
>
> ## Q4: Questions about sensibility of hyperparameters.
> We appreciate you asking for insights from our experience. Our core finding is that the high quality of data provides significant robustness and generalization benefits, diminishing the overall sensitivity to hyperparameter tuning compared to training on less-detailed datasets. The primary challenge was not optimization sensitivity, but achieving modality balance, which we solved through the two-stage curriculum.
>
> ------------
>
> ## Conclusion
>
> We hope these responses fully address your minor points of curiosity. We sincerely hope you will consider increasing your rating and confidence. Thank you again for your time and expertise.

---

> > ### Comment · Reviewer_WAFR · 2025-11-24
> >
> > Thank you for the thorough rebuttal and revisions. My concerns have been fully addressed, and I believe the paper makes a meaningful contribution to the community. After reviewing the other reviewers' comments as well, it appears that the remaining concerns have been adequately resolved or are minor relative to the strengths of the work, and do not affect the key considerations for acceptance. I support accepting this paper.

---

### Official Review · Reviewer_Mhp2 · 2025-11-01

**Soundness:** 1
**Presentation:** 4
**Contribution:** 4
**Rating:** 2
**Confidence:** 5

**Summary:**

This submission introduces Omni-Detective, an agentic data pipeline for generating highly detailed yet low-hallucination multimodal captions via iterative tool-calling and evidence verification.
Using this data, the authors train Audio-Captioner (audio-only) followed by Omni-Captioner (audio–visual), and claimed that they achieve state-of-the-art results (but this reviewer got some concerns and this matter should be discussed further).
To evaluate fine-grained multimodal perception, they also propose Omni-Cloze, a cloze-style benchmark covering audio, visual, and audio–visual tasks, offering stable and efficient assessment with strong alignment to human preference.
Overall, the work presents a whole stack, including data pipeline, models, and benchmark, for fine-grained, low-hallucination multimodal understanding.

This reviewer saw the contribution and potential of this submission to the community. However, this reviewer found that the comparative evaluations are conducted in unfair ways. This issue may nullify most of the claims made by comparing with the competing methods. This should be further discussed during the rebuttal.

**Strengths:**

- Conceptual clarity: Clearly articulates the "co-growth" issue between detail and hallucination and provides an actionable framework to mitigate it.

- Clear demonstration of the effectiveness of the Omni-Detective

- Comprehensive evaluation: Omni-Cloze provides stable, low-cost, human-aligned evaluation across modalities.

- Reproducibility: Extensive appendices with prompts, hyperparameters, and validation details enhance transparency.

Overall, the paper structure, motivation, contribution, and analyses are well implemented.

**Weaknesses:**

- Unfair comparison due to unfair training scheme: The comparison in Table 2 appears not fully fair, as the proposed Omni-Captioner is trained with the additional proposed large-scale, detailed captioning data (≈ 55k audio-only + 15k audio-visual samples) from VGGSound and FineVideo using a two-stage curriculum. In contrast, all baseline results are "obtained from their original papers" (as described in the caption of Table 2), meaning those models were applied without any chance to be fine-tuned with comparable detailed captioning datasets, but evaluated in their own (likely general or zero-shot) settings. Although the authors mentioned the proposed Captioners were applied in a zero-shot way, the zero-shot meanings are different (the proposed method is not fine-tuned on any training set related to the benchmark, but it still has an unfair advantage of fine-tuning on additional large-scale detailed captioning data, while the other competing methods do not).
Thus, the proposed model benefits from substantially more task-specific supervision, making the performance gains partly attributable to data scale rather than modeling improvements. The authors should clarify this mismatch or conduct controlled experiments with at least comparable training datasets, so that Table 2 clearly shows the effectiveness of the proposed dataset and training recipe. If there is no comparable dataset for the audio-only case, the authors should provide some baselines at least in an ablation way.

- Due to the above concern, all the claims made with the comparisons against the competing methods are deemed to be nullified. For example, in Table 3, even if the proposed method is additionally fine-tuned, the performance gaps against the other competing methods seem marginal.

- Scalability: While the design of Omni-Detective seems effective according to Fig. 6, it may be bulky and has a trade-off between the effectiveness and computational costs. The multi-step Omni-Detective pipeline may be computationally expensive for large-scale data generation, but no analysis and discussion have been reported.

**Questions:**

Please check the weakness for further discussion.

Personally, this reviewer would like the work, but the following critical concern hinders giving a positive opinion.
However, the unfair comparison is tightly entangled with the overall contribution claims. It's indeed unfortunate.
Thus, carefully addressing this reviewer's concern would be essential to re-evaluate the submission. Otherwise, the rate would not be changed.

---

> ### Author Response · Authors · 2025-11-23
> **General Response**
>
> Dear Reviewer Mhp2,
>
> Thank you for your thorough review and for recognizing the conceptual clarity, comprehensive evaluation, and contribution of our work to the community. We appreciate your positive remarks on the clarity of the "co-growth" issue and the effectiveness of Omni-Detective.
>
> We have carefully considered your concern regarding the fairness of the comparisons (Weakness 1 & 2). We first specifically discuss why video-SALMONN 2 is a fair baseline, and then we perform additional experiments and provided clarifications to fully address this concern.

---

> ### Author Response · Authors · 2025-11-23
> **Response 1**
>
> ## W1 & W2: Concern on baseline comparison
> You correctly pointed out that our models benefit from being fine-tuned on additional large-scale detailed captioning data generated by Omni-Detective, while baselines mostly report results from their original, general, or zero-shot settings. Our primary claim is that the Omni-Detective yields significantly higher-quality, less-hallucinatory detailed captions than prior methods, which, in turn, enables our models to achieve state-of-the-art results. To demonstrate that the superiority comes from the Omni-Detective rather than merely a larger quantity of data, we offer the following arguments and new, controlled **cross-ablation** and **self-ablation** studies:
>
> ### **1) The Strongest Fair Baseline: video-SALMONN 2**
> We specifically chose to compare against video-SALMONN 2 because its core focus, as indicated by its title ("Captioning-enhanced Audio-Visual Large Language Models"), is the direct enhancement of detailed captioning capabilities. This is **exactly the same task to ours**, and the results we cite from its original paper represent the current SOTA performance on the VDC and video-SALMONN-test detailed captioning benchmark. Our performance gains over this most relevant baseline are a strong indicator of our method's effectiveness.
>
> ### **2) Cross-ablation: comparing Omni-Detective data to other high-quality video datasets**
>
> To test the value of the Omni-Detective pipeline itself, we conducted a new controlled experiment:
> - Baseline Data: We collected **a larger amount of 20k detailed visual-only captions** from the publicly available LLaVA-Video-178K dataset, which is designed for video understanding.
> - Our Data: We used our smaller 15k data from Omni-Detective.
> - Model: We fine-tuned the same Qwen2.5-Omni-7B backbone using both datasets separately and evaluated their performance on the VDC (visual-only) benchmark.
>
> | **Data Source (Size)**   | **Acc% ($\uparrow$)** | **Score ($\uparrow$)**  |
> |--------------------------|---------------------------|----------------------------|
> | **Omni-Detective (15k)** | **55.02**                | **2.75**                   |
> | **LLaVA-Video (20k)**    | 52.15                    | 2.57                       |
>
> Our Omni-Detective data pipeline yields superior performance **even with a smaller dataset size (15k vs 20k)**. This indicates that the quality of data generated through our agentic verification and iterative refinement is more impactful than merely scaling up the number of samples from other high-quality sources, effectively validating the Omni-Detective's effectiveness.
>
> ### **3) Self-ablation: comparing Omni-Detective to conventional aspect-driven captioning**
>
> For the audio-only modality, where comparable detailed captioning datasets do not exist, we devised a self-ablation to compare Omni-Detective's output against a common alternative data generation strategy:
> - Baseline Data: We used the same 55k audio-only clips. Instead of running Omni-Detective, we used a high-performance MLLM (Gemini 2.5 Pro) with four aspect-specific prompts (Signal, Perception, Semantic, and Cultural layers, inspired by MMAR's taxonomy) to generate individual aspect-driven captions. These captions were then fused by GPT-4o into a single detailed caption. **This is a common pipeline in multimodal data curation and strong align with MMAR benchmark design.**
> - Our Data: 55k Detailed captions generated by Omni-Detective's iterative process.
> - Evaluation: Both generated datasets were used to fine-tune Qwen2.5-Omni-7B and evaluated on the MMAR cascade benchmark.
>
> | **Data Score** | **Sp% ($\uparrow$)** | **Mu% ($\uparrow$)** | **So% ($\uparrow$)** | **So-Mu% ($\uparrow$)** | **So-Sp% ($\uparrow$)** | **Mu-Sp% ($\uparrow$)** | **So-Mu-Sp% ($\uparrow$)** | **Avg.% ($\uparrow$)** |
> |---|---|---|---|---|---|---|---|---|
> | **Omni-Detective** | **55.15** | 46.12 | **68.37** | **63.64** | **61.01** | **67.07** | **66.67** | **59.80** |
> | **Captioning Combination** | 47.27 | **48.06** | 62.24 | **63.64** | 58.26 | 64.63 | 54.17 | 56.00 |
>
> Fine-tuning on Omni-Detective's data significantly outperforms the model trained on data generated by conventionally combining aspect-specific captions, **even though the aspects are specially designed to align with MMAR's taxonomy**. This further confirms that Omni-Detective is superior for capturing the complex, low-hallucination details for audio tasks.
>
> We believe these two new controlled ablation studies decisively address your concern by demonstrating that the gains are primarily attributable to the efficacy of the Omni-Detective and not simply to the scale (cross-ablation) and task-specific supervision (self-ablation) of the training data.

---

> ### Author Response · Authors · 2025-11-23
> **Response 2**
>
> ## W3: Scalability and computational costs
> We fully acknowledge your point that the multi-step Omni-Detective pipeline involves a trade-off between effectiveness and computational cost; it is fundamentally a computation-for-quality exchange, which we believe is necessary and worthwhile for achieving high-fidelity ground truth data.
>
> As an agentic data generation pipeline, precisely estimating the total computational cost (both time and resources) is inherently complex for several reasons:
> 1. **Adaptive Step Budget**: The agent adaptively schedules its investigation steps, and the required number of turns is variable depending on the content complexity of the video clip (the agent may early stop or use its full budget of steps).
> 2. **Tools/APIs Variability**: Specialist tools are called via API or local deployment. The latency and throughput are dependent on external service loads and concurrency settings, making stable time estimation difficult.
>
> For our setting (as noted in Appendix A.1.1), the average generation time per high-quality data sample is approximately 12 minutes (with a concurrency level of 100), resulting in around 7 seconds per sample. We consider this a highly efficient cost for producing a dataset of unprecedented quality and detail. This upfront investment in data quality significantly improves the performance frontier for models trained on this data, as shown in our results.
>
> ------------
>
> ## Conclusion
>
> We sincerely hope that the detailed clarifications and new controlled experimental results have fully addressed your concerns regarding the fairness of comparison and the data generation methodology. We look forward to your re-evaluation of our submission. Thank you again for your time and expertise.

---

> ### Author Response · Authors · 2025-11-27
> **Promote Discussion**
>
> Dear Reviewer Mhp2,
>
> We sincerely appreciate the time and expertise you invested in reviewing our paper. We have carefully addressed all concerns raised in the reviews and have provided detailed point-by-point responses. As the discussion period is now underway, we would be grateful if you could review our responses and post your comments. We are committed to engaging to enhance the quality of our work.
>
> Thanks again for your continued attention.
>
> Best regards,
>
> The Authors

---

### Author Response · Authors · 2025-11-26
**Promote Discussion**

Dear Reviewers,

We sincerely appreciate the time and expertise you invested in reviewing our paper.
We have carefully addressed all concerns raised in the reviews and have provided detailed point-by-point responses.
As the discussion period is now underway, we would be grateful if you could review our responses and post your comments.
We are committed to engaging to enhance the quality of our work.

Thanks again for your continued attention.

Best regards,

The Authors

---

### Meta-Review · Area_Chair_2XKW · 2026-01-06

**Summary:**

Reviewers raised concerns about baseline fairness, missing experimental details, and evaluation clarity. Key issues regarding baseline comparisons (e.g., reviewers Mhp2, iKJk, jZBM) and missing experiments or details (reviewer 1S9o) were addressed in the rebuttal through additional ablation studies and expanded experimental results. Overall, the rebuttal resolved the primary concerns, supporting the suggested decision of acceptance.

**Reviewer Concerns:**

Reviewer Mhp2 primarily raised concerns regarding potentially unfair baseline comparisons. This issue was addressed in the rebuttal through the addition of cross-ablation and self-ablation experiments.

Reviewer 1S9o expressed concerns about missing methodological details and experiments; most of these points were addressed in the rebuttal. While the authors’ claim regarding LoRA lacked strong supporting evidence, this issue is relatively minor in the context of the paper.

 Reviewers WAFR and AErd provided high overall ratings, and their concerns were limited to minor issues.

**Reviewer Scores:**

Reviewer Mhp2 and Reviewer 1S9o would likely increase their scores to 6 or above following the rebuttal, while the remaining reviewers are expected to keep their scores unchanged.

---

### Decision · Program_Chairs · 2026-01-26

Accept (Poster)